# The effect of COVID certificates on vaccine uptake, health outcomes, and the economy

Miquel Oliu-Barton[1,2 ✉], Bary S. R. Pradelski [3 ✉], Nicolas Woloszko [4 ✉], Lionel Guetta-Jeanrenaud[2], Philippe Aghion[5], Patrick Artus[6], Arnaud Fontanet[7], Philippe Martin[8] & Guntram B. Wolff [2,9]

In the COVID-19 pandemic many countries required COVID certificates, proving vaccination, recovery, or a recent negative test, to access public and private venues. We estimate their effect on vaccine uptake for France, Germany, and Italy using counterfactuals constructed via innovation diffusion theory. The announcement of COVID certificates during summer 2021 were associated – although causality cannot be directly inferred – with increased vaccine uptake in France of 13.0 (95% CI 9.7–14.9) percentage points (p.p.) of the total population until the end of the year, in Germany 6.2 (2.6–6.9) p.p., and in Italy 9.7 (5.4–12.3) p.p. Based on these estimates, an additional 3979 (3453–4298) deaths in France, 1133 (−312–1358) in Germany, and 1331 (502–1794) in Italy were averted; and gross domestic product (GDP) losses of €6.0 (5.9–6.1) billion in France, €1.4 (1.3–1.5) billion in Germany, and €2.1 (2.0–2.2) billion in Italy were prevented. Notably, in France, the application of COVID certificates averted high intensive care unit occupancy levels where prior lockdowns were instated.

[1] Université Paris-Dauphine, Paris, France. [2] Bruegel, Belgium. [3] French National Centre for Scientific Research (CNRS), Paris, France. [4] Organisation for Economic Co-operation and Development (OECD), Paris, France. [5] Collège de France and INSEAD, Paris, France. [6] Natixis and Paris School of Economics, Paris, France. [7] Institut Pasteur and Conservatoire National des Arts et Métiers, Paris, France. [8] Sciences Po and CEPR, Paris, France. [9] Solvay Brussels School, Université libre de Bruxelles, Brussels, Belgium. ✉email: miquel.oliu.barton@normalesup.org; bary.pradelski@cnrs.fr; nicolas.woloszko@oecd.org

The COVID-19 pandemic has forced many governments to implement previously unthinkable policies[1,2]. Initially, while some countries aimed to eliminate the virus, others aimed to slow its spread to protect health systems and to gain time until vaccines or treatment became widely available[3–5]. Public health measures intended to reduce transmissions have included public venue closures, limitations on social contacts, and travel restrictions. These measures are informed by mathematical simulations[6–8] and by analyses estimating their causal effects on the dynamics of the epidemic[9,10]. COVID certificates—certifying vaccination, recovery, or a recent negative test—enable, through digitisation, targeted interventions dependent on an individual's risk of transmitting the virus or experiencing a severe form of the disease[11]. As with other policy choices, the use of COVID certificates has often been questioned for ethical and political reasons[12,13], while advocates have focused on the potential to secure social interactions[14] and only recently increased vaccine uptake has been considered[15–19]. We argue that the incentive effect of COVID certificates on vaccine uptake may be most critical and has averted adverse health and economic outcomes. As vaccines are increasingly available, hesitancy and refusal to be vaccinated have become the main obstacles to high vaccine coverage in many parts of the world[20–22]. Historically, policy-makers have considered several options to increase vaccine uptake, ranging from communication and outreach strategies to monetary (dis)incentives and mandates[23]. COVID certificates have emerged during the pandemic as a new tool to spur vaccination uptake and they require further investigation.

In Europe, the use of COVID certificates for travel was agreed upon within the European Union (EU) in June 2021[24]. Several member states, including France, Germany, and Italy, subsequently adopted COVID certificates for many domestic activities. We focus on these countries as they introduced this tool at similar times to regulate entry to public venues, restaurants, cafes, bars, shops, etc. (on 12 July 2021, 10 August 2021, and 22 July 2021). While COVID certificates were required throughout France and Italy, in Germany they were only required in regions where the seven-day incidence was above 35 per 100,000 (see Methods A for detailed lists of the regulations in each country). The three countries further have comparable *per-capita* vaccine supply[25], demographics, health infrastructure, and economies (see Supplementary Table 1, Appendix). Our objective is to measure how much the widespread use of COVID certificates incentivised vaccine uptake, reduced adverse health outcomes, and strengthened the economy. This study may thus help to inform decision-making on whether, when, and how to employ COVID certificates.

## Results

**COVID certificates spur vaccination**. We estimate the COVID certificate's contribution to vaccine uptake in France, Germany, and Italy by constructing counterfactuals (i.e., by modelling vaccine uptake without this intervention), using innovation diffusion theory[26]. Innovation diffusion theory was introduced to model how new ideas and technologies spread[26] and, among other applications, has been used to study the uptake of medical innovations[27], in particular vaccines[28]. The theory captures the way in which an innovation—the vaccine—is gradually taken up by a population, with early adopters subsequently joined by followers[29]. Mathematically, the model relies on growth theory with capacity limits and two critical parameters. On the one hand, the "coefficient of innovation" is the instantaneous rate at which a non-vaccinated person opts to get vaccinated, independent of how many people are already vaccinated. On the other hand, the "coefficient of imitation" is the rate at which a non-vaccinated person is influenced by the share of vaccinated people in their decision to get vaccinated. This micro-founded model has been widely used due to its tractability and interpretability[30], is robust to capacity constraints, but requires the assumption of no exogeneous shocks (Methods B).

*Findings.* The effects of COVID certificates on vaccine uptake turned out to be sizeable (Fig. 1). On the day of their announcements, in France, 53.8% of the population had received at least one dose of a COVID-19 vaccine, in Germany 62.5%, and in Italy 61.6%. By the end of 2021, the first-dose vaccine uptake had risen to 78.2%, 73.5%, and 80.1%, respectively. How much of this increase can be attributed to COVID certificates? To answer this question, we estimate the counterfactual without COVID certificates to be approximately 65.2% in France, 67.3% in Germany, and 70.4% in Italy (Methods B). More precisely, we attribute 13.0 (95% CI 9.7–14.9) percentage points (p.p.) of vaccine uptake for France, 6.2 (2.6–6.9) p.p. for Germany, and 9.7 (5.4–12.3) p.p. for Italy to the incentives created by COVID certificates. All three countries further extended the use of COVID certificates in the months following their initial announcement, ranging from their requirement in workplaces to the integration of a booster dose. Our estimates include the incentives created by these additional extensions. The overall effect is significant in France and Italy, but only from end of November 2021 onwards in Germany, when the use of COVID certificates was extended to workplaces (Fig. 1). Moreover, in France and Italy –where age-dependent vaccine uptake statistics were available– we find that the impact was larger among the younger population. Last, we did not find sizeable spillover effects between the announcements of Covid certificates in France, Germany, and Italy (see Methods B).

Next, we estimate the effect of COVID certificates on vaccine uptake by splitting the population into over and under 60 years of age. This is particularly relevant as health outcomes are generally more severe for older people. By the end of 2021, we attribute for France 8.9 (8.0–9.4) p.p. to the incentives created by COVID certificates among the population over 60 years old, and for Italy 4.4 (2.9–5.2) p.p. (Methods B). Thus, the impact is significant among the older population, although it is even larger among the younger population. Germany did not publish age-dependent vaccine uptake statistics until mid-September 2021, so we cannot build a counterfactual for vaccine uptake among the older population in Germany.

Our results are supported by the well-established econometric method of synthetic control[31]. We construct counterfactuals for each treated country based on a weighted average of countries that did not implement the COVID certificate and find consistent trajectories –that is, falling in the 95% confidence interval of the innovation diffusion model– for the time period where this method is feasible, i.e., until the end of September 2021 (Methods B). As synthetic control is robust to shocks that are common to all countries, this suggests that around the period of the introduction of COVID certificates exogenous shocks, such as the rise of the Delta variant from late June, did not crucially pollute our estimates. Thus, their exclusion in the innovation diffusion model is appropriate. Synthetic control offers a valuable robustness check but has limitations in our context. The method requires a sufficiently large control group, which is infeasible as more and more countries adopted COVID certificates in fall 2021, hence our choice to use an alternative principal method. Further, synthetic control requires that the countries in the control group are not affected by interventions in other countries, which is questionable given the interdependence of COVID-related policies, and cross-border interactions.

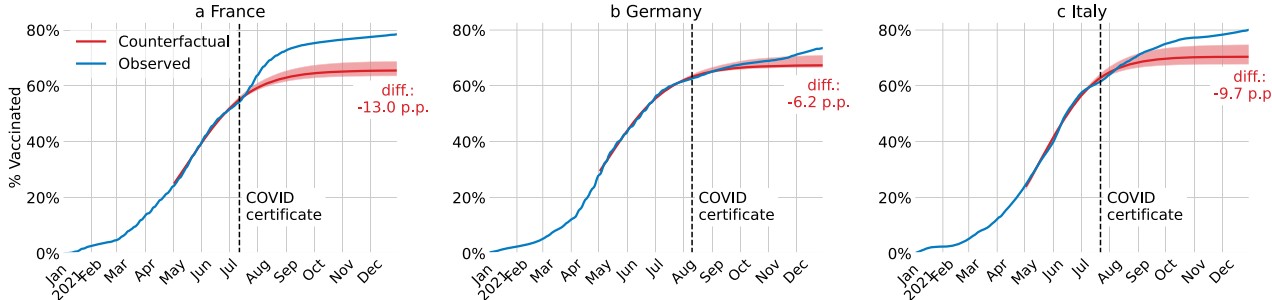

**Fig. 1 Estimated vaccine uptake with and without COVID certificates.** For France (**a**), Germany (**b**), and Italy (**c**), cumulative share of the population who received at least one COVID-19 vaccine dose in the actual intervention deployment (blue) and in the no-intervention counterfactual scenario (red). The red shaded area is the 95% confidence interval centred around the main estimate. The black dashed vertical line is the date of the announcement of the COVID certificate.

Our results are consistent but overall more substantial than predicted by survey-based estimates[32], and are in line with studies analysing the immediate period after the intervention in various countries using cross-country or state comparisons using different methods[15–19]. In particular, using synthetic control, Mills and Rüttenauer[15] estimate the effect of COVID certificates on vaccine uptake 20 days before (as a proxy for the date of their announcement) and 40 days after their implementation at 12.8 (8.8–18.8) p.p. of the entire population for France, 2.5 (−4.3–6.7) p.p. for Germany, and 6.6 (2.6–12.7) p.p. for Italy, with a larger effect among the younger population. Using time series methods, Karaivanov et al.[16] estimate the effect of COVID certificates on vaccine uptake from their announcement until 31 October 2021 at 8 p.p. of the entire population for France, 4.7 p.p. for Germany, and 12.1 p.p. for Italy. By contrast, we estimate the effect from the announcement dates until the end of 2021 using innovation diffusion theory.

**Impact of COVID certificates on health outcomes**. The effectiveness of COVID-19 vaccines against hospitalisation, intensive care unit (ICU) admission, and death has been well documented, including for the Delta variant, which was prevalent throughout the period of study. We estimate the average effectiveness, considering the various vaccines and waning immunity, at 81% after one dose and 92% after two doses by considering lower bounds from the medical literature (Methods C)[33–37]. We focus on the direct protection provided by vaccines, but omit the contribution of vaccines to reducing overall transmission and the fact that COVID certificates may alter epidemic dynamics through, for example, behaviour changes or different patterns of increasing natural immunity.

To estimate the impact of vaccine uptake on health outcomes, we construct counterfactuals for second-dose vaccine uptake by assuming the same ratio between second and first dose uptake (with a three-week lag) for the counterfactual and realised scenarios (Methods B). Booster uptake does not factor in our model, as individuals who were not vaccinated before the announcement of the COVID certificate were not eligible to receive a booster during 2021 (Methods B). We consider age-stratified uptake estimates when available; in particular, this is the case for France and Italy for deaths, and for France for hospital admissions (Methods C).

*Findings.* We estimate the number of hospital admissions and deaths that would have occurred from the announcement of COVID certificates until the end of 2021 (Methods C and Fig. 2). In France, an additional 32,065 (26,566–35,306) hospital admissions would have occurred, in Germany 5229 (−1774–6822), and

in Italy 8735 (2999–12,261). Additional deaths in France would have been 3979 (3453–4298), in Germany 1133 (−312 to 1358), and in Italy 1331 (502–1794). Thus, from the introduction of COVID certificates until the end of 2021, the expected number of hospital admissions (and deaths) would have been 31.3% (31.7%) higher in France, 5.0% (5.6%) higher in Germany, and 15.5% (14.0%) higher in Italy. Notably, the impact of additional vaccine uptake compounds over time, and while the effect is significant for France and Italy over the entire period, it only becomes significant for Germany by the end of November. In the last week of 2021, without the accumulated difference in vaccine uptake, there would have been approximately 46% (49%) more hospital admissions (deaths) in France, 14% (11%) more in Germany, and 29% (26%) more in Italy (Fig. 2).

**Impact of COVID certificates on economic activity**. The COVID-19 pandemic has had a large negative impact on the economy[38,39]. COVID certificates may spur economic recovery in the short run, as newly vaccinated people can safely resume in-person economic activities, including working on-site, and consuming goods as well as services in brick-and-mortar businesses (direct effect). Furthermore, an indirect effect results from avoiding restrictions, through public health measures, on social, education, and economic activities. Here, we conduct a quantitative analysis of the overall economic effect of COVID certificates based on the weekly GDP estimates provided by the OECD Weekly Tracker[40]. Resorting to a high-frequency indicator of economic activity is necessary to exploit the weekly variations in vaccination rates to identify the effect of vaccine uptake on economic activity. This paper innovates compared to previous analyses[41], as the use of a high-frequency GDP proxy allows quantitative estimates of the economic impact of variations in vaccination rates.

The impact of COVID certificates on the economy is modelled through its effect on vaccine uptake and the elasticity of the latter to weekly GDP, using data from all OECD and G20 member countries as used by the OECD Weekly Tracker (Methods D). The average effect of vaccination on GDP is estimated using a closed-form model in which weekly GDP is regressed on first-dose vaccination, lagged by a month to account for the time between first and second dose and time to full effectiveness. In addition, health outcomes, also lagged by a month, are controlled for, as they may be confounding factors influencing both vaccine uptake and GDP. Furthermore, we control for vaccination and health outcomes of the main trade partners to avoid the possible confounding effect of trade and other economic spillovers[41], and for average weekly temperature, which influences virus diffusion[42]. Finally, we add week and country fixed effects to

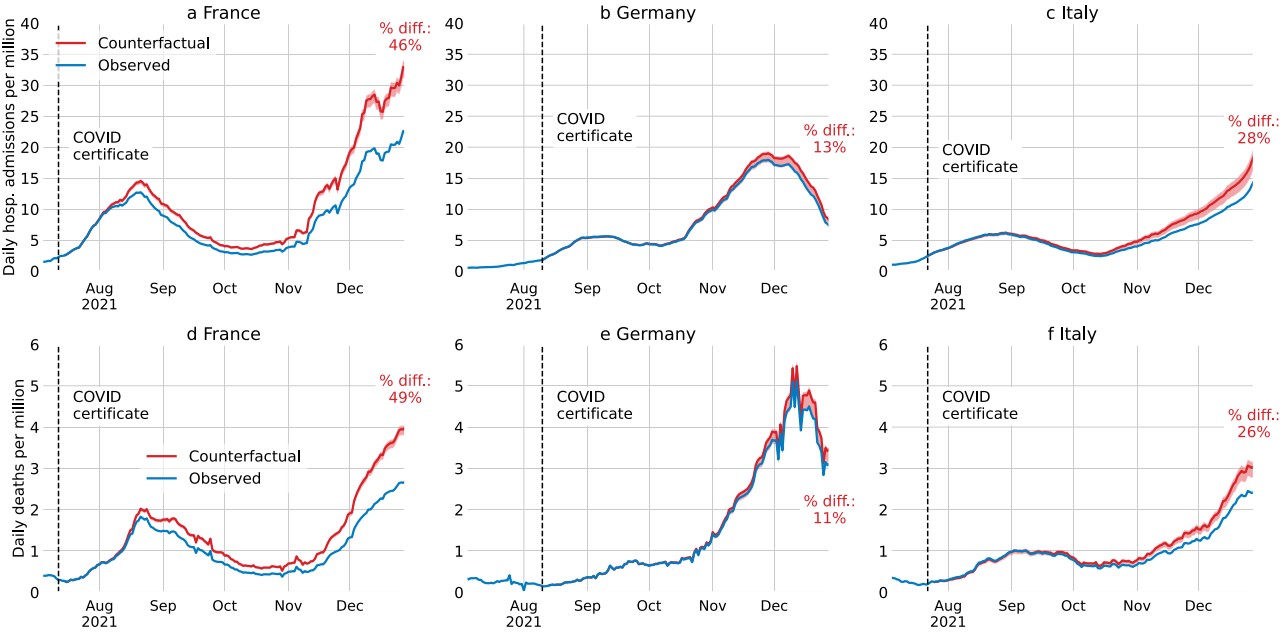

**Fig. 2 Estimated hospital admissions and deaths with and without COVID certificates.** For France (**a** and **d**), Germany (**b** and **e**), and Italy (**c** and **f**), daily hospital admissions (top row) and deaths (bottom row) per million (7-day rolling average) in the actual intervention deployment (blue) and in the no-intervention counterfactual scenario (red). The red shaded area is the 95% confidence interval centred around the main estimate. The daily death counterfactuals for France and Italy, and the daily hospital admissions counterfactual for France are computed using an age-stratified model. The other counterfactuals are not based on age-stratified models due to unavailable data. The black dashed vertical line is the date of the announcement of the COVID certificate.

control for any common seasonal effects and any country-specific but time-invariant effects, such as demographic or geographical characteristics.

*Findings.* The average effect of a 1 p.p. increase in the share of vaccinated people on weekly GDP one month later is 0.052 (0.033–0.070) p.p. A complete vaccination roll-out would thus drive GDP up by 5.2 p.p., which corresponds to approximately 85% of the loss observed in 2020. We estimate counterfactual weekly GDP trajectories for France, Germany, and Italy based on estimated counterfactual vaccine uptake (Fig. 3). By the end of 2021, without the policy intervention, weekly GDP would have been 0.6 (0.5–0.9) % lower in France, 0.3 (0.1–0.4) % lower in Germany, and 0.5 (0.3–0.7) % lower in Italy, amounting to GDP losses across the second half of 2021 of €6.0 (5.9–6.1) billion in France, €1.4 (1.3–1.5) billion in Germany, and €2.1 (2.0–2.2) billion in Italy (Methods D).

We corroborate our findings with robustness checks regarding the statistical method, the choice of lag between vaccination and GDP, the modelling assumption that the vaccine-GDP relationship did not vary substantially across the considered time period, and alternative dependent variables, namely the Google Maps mobility index and official quarterly GDP (Methods D).

**COVID certificates may have prevented lockdowns.** By increasing vaccine uptake, COVID certificates reduced the number of patients in ICUs and thus contributed to reducing the likelihood of stricter public measures, including lockdowns. While such decisions are ultimately made by governments, their anticipation and perceived uncertainty are harmful to the economy, also in the mid- and long-term[43]. It is thus instructive to consider the evolution of ICU patients over time and to use levels of previous lockdowns as benchmarks. We exclude the first lockdowns from the analysis, as they represent unrealistic benchmarks for future government action due to unprecedented

uncertainty. In France, the number of COVID-19 patients in ICUs per million was 44.9 when the second lockdown was announced (28 October 2020) and 74.8 when the third lockdown was announced (31 March 2021). In Germany, it was 54.3 (second lockdown, 13 December 2020), in Italy 53.0 (second lockdown, 12 December 2020) and 60.2 (third lockdown, 27 March 2021).

*Findings.* By the end of 2021, in France, the number of COVID-19 patients in ICUs was 52.4 per million. We estimate it would have been 76.1 (72.4–78.3) without the introduction of COVID certificates, i.e., for the central estimate, an increase of 45% (Methods C). The policy intervention may thus have been instrumental in preventing the high pressure on ICUs that prompted previous lockdowns (Fig. 4). By contrast, the additional vaccine uptake in Germany was not sufficient to avert high pressure on ICUs. Consequently, more stringent measures were adopted. Finally, COVID certificates did not play a decisive role in Italy during the period under investigation, as the pressure on ICUs would have remained at low levels even without the policy intervention.

**Discussion**

COVID certificates were associated with a sizeable, robust positive effect on vaccination rates, health outcomes, and the economy in France, Germany (albeit only significantly towards the end of 2021), and Italy. In our analysis, we aimed to make prudent assumptions on model inputs such as vaccine effectiveness and lags between infection and health outcomes. Nevertheless, our analysis does not allow to directly infer causality and relies on the estimated increase in vaccine uptake, while not consider how COVID certificates may have altered epidemic dynamics as well as influenced other policy choices. Further study in this direction is required. In addition, we evaluated the temporary effect of the incentives created by COVID certificates, but effects may be

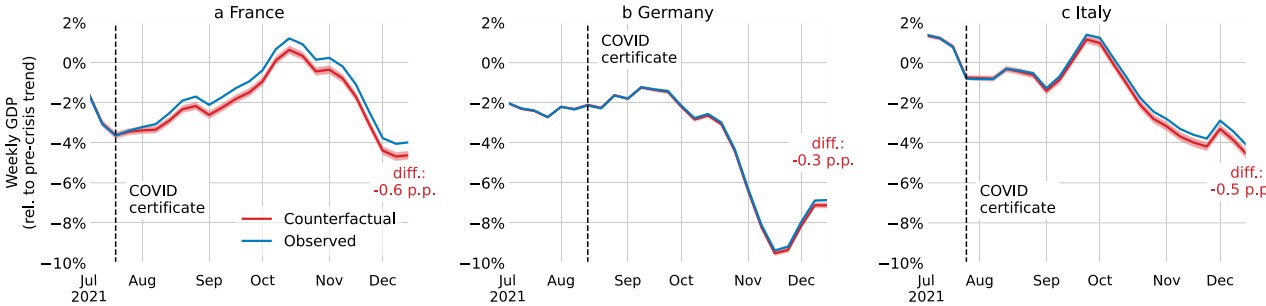

**Fig. 3 Estimated weekly GDP with and without COVID certificates.** For France (**a**), Germany (**b**), and Italy (**c**), weekly GDP (3-week rolling average) in the actual intervention deployment (blue) and in the no-intervention counterfactual scenario (red). The red shaded area is the 95% confidence interval centred around the main estimate. The black dashed vertical line is the date of the announcement of the COVID certificate.

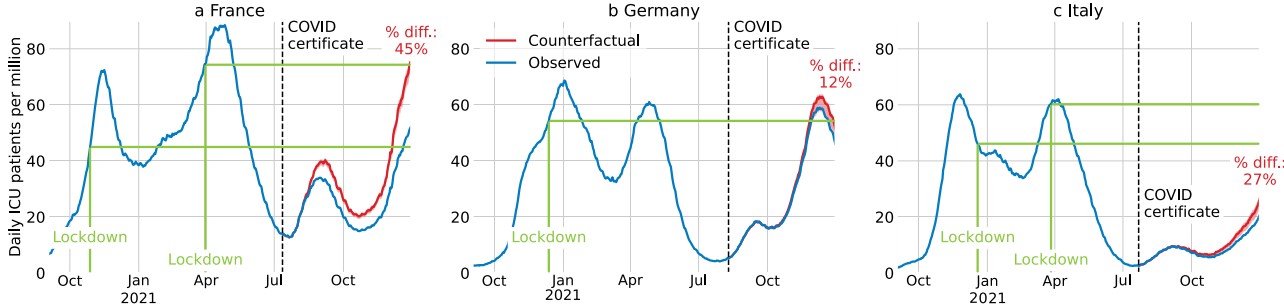

**Fig. 4 Estimated ICU patients with and without COVID certificates.** For France (**a**), Germany (**b**), and Italy (**c**), daily COVID-19 patients in intensive care units (ICUs) per million (7-day rolling average) in the actual intervention deployment (blue) and in the no-intervention counterfactual scenario (red). The red shaded area is the 95% confidence interval centred around the main estimate. The counterfactual for France is based on an age-stratified model. The other counterfactuals are not based on age-stratified models due to unavailable data. Green lines indicate levels at which previous lockdowns were instated. The black dotted vertical line is the date of the announcement of the COVID certificate.

lasting, as parts of the population might have remained unvaccinated without the intervention.

Even though COVID certificates were introduced in similar contexts—i.e., the three largest EU countries, at a time when vaccination campaigns were slowing down, and infections were increasing rapidly –the magnitude of the impact varies significantly from one country to the other and reversed previous trends associated with vaccine hesitancy, and lack of trust in science and government[20–22]. Understanding these differences deserves attention. Factors could be the various ways in which COVID certificates were announced and the extent of their use. For example, the announcements in France and Italy were particularly striking, as they were made by the central governments and backed by clear and consistent communication, with COVID certificates being required in most public venues all over the country. By contrast, COVID certificates were introduced more gradually in Germany with different rules and enforcement levels across its federal states and depending on the local incidence. Another consideration is the timing of the introduction, notably Germany's later announcement might thus explain the smaller effect on vaccine uptake. Further studies should complement our work by assessing the broader effect of COVID certificates on the development of the epidemic. Additionally, long-term social and economic costs and benefits need to be considered as more data will become available.

COVID certificates appear to be an attractive, more inclusive alternative to vaccine mandates, focusing on the added benefits of getting vaccinated or tested rather than on punitive measures for not doing so. As countries grapple with the highly contagious Omicron variant, COVID certificates might play a decisive role in increasing and maintaining vaccine-induced protection. Nevertheless, governments' policy decisions on COVID certificates should also consider additional factors, including supply of

vaccines and tests, political trust, and accessibility for marginalised groups, in order not to threaten social cohesion or exacerbate already existing inequities[12,44,45]. Finally, international coordination and mutual acceptance of COVID certificates are crucial to prevent deepening the divide between different regions [11].

## Methods
**Data**. All data were retrieved in the first week of January 2022.

*Demographics, health infrastructure, and economic indicators*. The indicators in Supplementary Table 1 (Appendix) support the claim that France, Germany, and Italy are similar in terms of demography, health infrastructure, and their economies.

*Health data*. For all OECD and EU countries, the share of the population who received one dose of a COVID-19 vaccine, two doses of a COVID-19 vaccine, hospital admissions per 1 million, daily ICU patients per 1 million, daily deaths per 1 million, and population estimates have been retrieved from Our World In Data[46]. For France age-stratified data on hospital admissions, ICU patients, and deaths was retrieved from official government sources (https://www.data.gouv.fr/fr/datasets/synthese-des-indicateurs-de-suivi-de-lepidemie-covid-19/) and deaths outside hospitals from the French Institute for Demographic Studies (INED) (https://dc-covid.site.ined.fr/en/data/france/). For Italy age-stratified data on deaths was also retrieved from INED (https://dc-covid.site.ined.fr/en/data/italy/).

Age-stratified vaccine uptake statistics for France and Italy were both retrieved from the European Centre of Disease Prevention and Control (note that such data is not available for Germany) (https://vaccinetracker.ecdc.europa.eu/public/extensions/COVID-19/vaccine-tracker.html#age-group-tab).

The share of different vaccines used until the end of 2021 in France, Germany, and Italy (made by BioNTech/Pfizer, Moderna, AstraZeneca, Janssen Pharmaceutica NV) have been retrieved from the official government sources (France: https://covidtracker.fr/vaccintracker/, Germany: https://impfdashboard.de/, Italy: https://www.governo.it/it/cscovid19/report-vaccini/).

*OECD Weekly Tracker*. The OECD Weekly Tracker (short 'Weekly Tracker') provides weekly estimates of economic activity based on Google Trends data and

**Fig. 5 The OECD weekly tracker. a** OECD Weekly Tracker for France, Germany, and Italy[40]. **b** Schematic depiction of the OECD Weekly Tracker that is providing a proxy of weekly GDP relative to the pre-crisis trend based on Google Trends search data and machine learning.

performs well across the 46 OECD and G20 countries in forecast simulations. The Tracker's methodology[40] relies on a machine learning algorithm, which extracts signals from search intensities related to approximately 250 categories of search keywords to infer a timely picture of the economy. It is trained on official GDP series to predict weekly GDP from the weekly Google Trends series. It provides estimates of weekly GDP relative to the pre-crisis trend.

The Tracker is based on several Google Trends variables that were hand-picked to cover a wide range of aspects of economic activity. Importantly, for our analysis, the Tracker only uses search behaviour on economic variables and not health variables. Data about search behaviours can be informative about consumption (e.g., related to searches for "vehicles", "household appliances"), labour markets (e.g., "unemployment benefits"), housing (e.g., "real estate agency", "mortgage"), business services (e.g., "venture capital", "bankruptcy"), industrial activity (e.g., "maritime transport", "agricultural equipment") and economic sentiment (e.g., "recession"), and poverty (e.g., "food bank"). Signals about multiple facets of the economy can be aggregated to infer a timely picture of the macroeconomy.

The relationship between the search volume indices and GDP, $f$, is learnt at the quarterly frequency using official quarterly GDP series and quarterly aggregates of the search indices. It is then used to disaggregate GDP growth at the weekly frequency by applying $f$ to the weekly search indices. The relationship between Google Trends variables and GDP growth is fitted using a neural network. It is trained using a dataset comprising the whole panel of observations from 46 countries.

The Tracker measures the percentage difference in GDP relative to a pandemic-free counterfactual, where the counterfactual is taken to be the OECD Economic Outlook projection published in November 2019[47]. Formally, the Tracker is defined as

$$T_w = \frac{y_w}{x_w} - 1, \qquad (1)$$

where $y_w$ is weekly GDP in week $w$, and $x_w$ is weekly GDP in a no-COVID counterfactual, proxied by a twelfth of quarterly GDP projected by the OECD Economic Outlook prior to the crisis. The Tracker thus measures weekly GDP relative to the pre-crisis trend (Fig. 5b).

Since it was released in December 2020, the Tracker has shed light on major policy issues related to the economic impact of lockdowns, infection waves, vaccination campaigns, and economic policy responses. The high-frequency nature of the Tracker makes it a relevant tool to assess the impact of COVID-19 policy responses. The Tracker series for France, Germany, and Italy are shown in Fig. 5a.

The OECD Weekly Tracker's accuracy was assessed using pseudo-real time forecast simulations[40]. On average across 46 countries over the period 2008Q1-2020Q4, it has a Root Mean Squared Error (RMSE) that is 17% lower than an autoregressive model that just uses lags of year-on-year GDP growth. The underlying model captures a sizeable share of business cycle variations, including the time around the global financial crisis (when the available data for training the algorithm was much smaller) and the euro area sovereign debt crisis. Its RMSE is on average 8% lower than an autoregressive model in 2008-10 and 41% lower in 2020. The timing of the downturn and subsequent rebound is well captured by the model, although the full magnitude of the negative shock in the second quarter of 2020 is typically under-estimated, given its unprecedented scale. The mean absolute error in predicting year-on-year GDP growth in the first (resp. second) quarter was 2.42 (resp. 3.86) p.p., compared with actual falls in GDP for the median country of 0.12% (resp. 10.4%). Note that the error is larger when the fall is very large, namely of a magnitude unseen before. The tracker thus provides a useful tool for real-time narrative analysis on a weekly basis, although it does not on average outperform models based on more standard variables, once these are eventually released.

*Vaccine acceptance.* Is taken from the University of Maryland Social Data Science Centre "Global COVID-19 Trends and Impact Survey" in partnership with Facebook (https://jpsm.umd.edu/research/global-covid-19-trends-and-impact-survey%2C-partnership-facebook). The survey is administered to a representative sample of Facebook users daily and includes questions on symptoms, social distancing behaviour, vaccine acceptance, mental health issues, and financial constraints. The acceptance rates used in this paper are built as the quarterly average of the proportion of respondents that said to 'definitely' or 'probably' choosing to get

vaccinated if a COVID-19 vaccine was offered to them over the first quarter of 2021. For the United States, acceptance rates were complemented using data from the Johns Hopkins Centre for Communication Programs. The acceptance rate for Malta was imputed using the median across countries.

*Mobility.* A mobility index is built from the Google Mobility reports (https://www.google.com/covid19/mobility/), which document mobility per type of destination relative to the pre-crisis levels at a daily frequency. The mobility index used in this paper is the simple average of mobility towards workplaces and places of retail and recreation.

*Temperature.* Daily temperature series for the 46 OECD and G20 countries across 2020 and 2021 were collected from the National Oceanic and Atmospheric Administration's National Centres for Environmental Information (https://www.ncei.noaa.gov/). The Global Historical Climatology Network daily (GHCNd) provides daily climate summaries from land surface stations across the globe. Temperature data for each station were averaged at the country level.

*Policy interventions.* For each country, we consider the date when COVID certificates for day-to-day use were announced, namely, 12 July 2021 for France, 10 August 2021 for Germany, and 22 July 2021 for Italy. See Supplementary Table 2 (Appendix) for details of the corresponding regulations.

*OECD and EU countries that announced the use of COVID certificates before 22 September 2021.* Austria, Canada, Chile, Colombia, Croatia, Cyprus, Denmark, Estonia, Finland, France, Germany, Greece, Ireland, Israel, Italy, Latvia, Lithuania, Luxembourg, Mexico, Portugal, Romania, Slovakia, Slovenia, Spain, Switzerland, the United States.

*Remaining OECD and EU countries.* Australia, Belgium, Bulgaria, Costa Rica, Czech Republic, Hungary, Iceland, Japan, Malta, the Netherlands, New Zealand, Norway, Poland, South Korea, Sweden, Turkey, the United Kingdom.

*Donor pool countries for synthetic control.* Australia, Belgium, Czech Republic, Hungary, Japan, Malta, the Netherlands, New Zealand, Norway, Poland, South Korea, Sweden, Turkey, the United Kingdom. (Costa Rica, and Iceland have been removed from the donor pool due to lack of data for covariates. Bulgaria has been removed from the donor pool due to lack of vaccination coverage data over most of the analysed period.)

See Supplementary Table 3 (Appendix) for information on all OECD and EU countries regarding the implementation of COVID certificates.

**Estimation of vaccine uptake.** The impact of COVID certificates on vaccination uptake is estimated using *innovation diffusion theory* and supported by the *synthetic control* method.

Innovation diffusion theory[26–30] attempts to formalise the way in which an innovation is gradually taken up by a population, where early adopters are then joined by followers. Every individual has their own, heterogeneous, threshold at which they decide to adopt the innovation. The model relies on growth models with capacity limits, i.e., logistic curves, positing that thresholds are distributed accordingly. In our context, vaccines are the innovation that every (eligible) person may choose to adopt.

Denote by $t_0$ the date when the vaccine is introduced and by $x(t) \in [0,1]$ the cumulative fraction of the population who has received at least one dose on day $t$. Thus, by assumption, $x(t) \in [0,1]$ for all $t$, the function $x(t)$ is nondecreasing, and $x(t)=0$ for all $t \leq t_0$. The innovation diffusion model depends on three additional parameters: $p > 0$ is the 'coefficient of innovation', i.e., the instantaneous rate at which a non-vaccinated person opts to get vaccinated, independently of how many people are already vaccinated; $q > 0$ is the 'coefficient of imitation', i.e., the rate at which a non-vaccinated person is influenced by the fraction of vaccinated people; and $0 < K \leq 1$ is the capacity, i.e., the fraction of the population that is eventually eligible and willing to get vaccinated.

Mathematically, the innovation diffusion model is described by the ordinary differential equation

$$x'(t) = \left(1 - \frac{x(t)}{K}\right)(p + qx(t)), \ t \geq t_0 \ and \ x'(t) = 0 \ elsewhere. \quad (2)$$

The unique solution to the latter differential equation is given by:

$$x(t) = K\frac{1 - e^{-(p+q)(t-t_0)}}{1 + \frac{q}{p}e^{-(p+q)(t-t_0)}}, \ t \geq t_0 \ and \ x(t) = 0 \ elsewhere. \quad (3)$$

Logistic functions model the diffusion of an innovation in the absence of major shocks, including supply shortages or policy interventions. While this is the case over the time period considered (i.e., date of announcement of COVID certificate to 31 December 2021), an extension to 2022 may be less appropriate due to the exogenous shock caused by the less severe Omicron variant becoming dominant in France, Germany, and Italy[48]. Regarding supply shortages and eligibility constraints, the early stages of the vaccine rollout were significantly impacted by supply constraints, which led most countries to give age-based priority. This effect could be captured by an extension of the original innovation diffusion model[49], but requires additional data, which is not available in our case (i.e., the fraction of the population willing to get vaccinated among the not-yet-eligible). Next to the imitation and innovation coefficients, this extended model would then add a 'word-of-mouth' coefficient that captures the influence of individuals willing to get vaccinated, not yet eligible, on the eligible population. When the word-of-mouth coefficient is assumed equal to the imitation coefficient, this model boils down to the original Bass model, thus lending further support to our modelling choice.

Innovation diffusion theory assumes constant parameters, which may be seen as a limitation. On the other hand, adding a time-dependent effect would result in a statistical model such as ordinary least squares (OLS), which does not have predictive power.

Parameters $t_0, p, q$, and $K$ are estimated using the least-square method to fit the data on vaccine uptake (see Table 1). The fit is computed over the 100 days prior to the announcement of a country's COVID certificate, when the majority of the adult population was eligible for vaccination, and then extended to the end of the year. For convenience, we define $t_0$ to be the number of days before or after 100 days prior to the announcement of the COVID certificate. Note that $t_0$ is not decisive for the estimation, as initial growth of the logistic function is near zero.

The fit is robust with respect to slightly longer or shorter fitting windows. On the other hand, when using a much shorter window (e.g., 60 days) estimates become very noisy with respect to the exact window length, likely because the fit does not appropriately pick up the curvature and inflection point of the logistic function. We use the function 'curve_fit' from Python's package 'scipy.optimize' over vaccine uptake and synthetic counterfactuals. We use block bootstrap to account for time dependence in the data with 1000 iterations and 30 non-

overlapping blocks[50]. The 95% confidence intervals are shown and reported throughout.

*Counterfactual vaccine uptakes.* For each country, denote by $V_t(v)$ for $v \in \{0, 1, 2\}$ the proportion of the population having received $v$ doses at time $t$. We do not consider $v \geq 3$ doses as individuals who were not vaccinated before the announcement of the COVID certificate were not eligible to receive a booster before early 2022; a counterfactual is thus not needed. (For example, in France, a person getting vaccinated on 12 July –the announcement date– could not get a booster until 2 January due to a 3-week gap between first and second dose, and 5 months gap for the booster).

For the first dose, the counterfactual is denoted by $\hat{V}_t(1)$, and is obtained from the estimation described above. Let $T_0$ denote the date when COVID certificates are announced in each country and $T_1$ be 31 December 2021. Then, $\hat{V}_t(1)$ is equal to $V_t(1)$ for all $t \leq T_0$ and is equal to the estimate obtained through the innovation diffusion model for all $t \in [T_0, T_1]$. To obtain a counterfactual for fully vaccinated individuals, $V_t(2)$, we assume the same ratio between first and second doses between the counterfactual and realised scenarios three weeks prior, i.e.:

$$\hat{V}_t(2) = \hat{V}_{t-21}(1) \cdot \frac{V_t(2)}{V_{t-21}(1)}. \quad (4)$$

Assuming the same ratio between second and first dose uptake (with a three-week lag, corresponding to the minimum required gap between first and second dose) for the counterfactual and realised scenarios is well-motivated as this ratio was not affected by the intervention (see Supplementary Fig. 1, Appendix). Finally, $\hat{V}_t(0) = 1 - \hat{V}_t(1) - \hat{V}_t(2)$.

For robustness, we estimated the impact of COVID certificates in Germany starting at the French announcement instead, and observed a slightly higher effect (i.e., 1.1 p.p. more by the end of the year). Assessing the impact of COVID certificates from the date it was announced in the country results in a more conservative estimate, as the potential impact of other countries' announcements is neglected.

*Age-stratified vaccine uptake.* To estimate age-stratified vaccine uptake, we consider the population aged 60 and over (60+) and the rest, separately. This is particularly relevant as health outcomes are generally more severe for older people. As age-stratified data are not available for Germany before mid-September 2021 it cannot be included in this analysis. We use the innovation diffusion model to construct a counterfactual for the 60+ group for France and Italy. For all vaccination statuses $v$, the realised vaccination uptake for 60+ is denoted by $V_t^{60+}(v)$, and the counterfactual is denoted by $\hat{V}_t^{60+}(v)$. To ensure consistency with our overall estimates, we set counterfactual vaccine uptake for the 59 years old and below as the difference between the overall and the 60+ estimates.

For the population aged 60 years and older, the fit is computed from the start of 2021 to the date of the announcement of the COVID certificate and then extended to the end of the year; the start date is chosen because by then the majority of the 60+ population was eligible for vaccination (Fig. 6). We use the function "curve_fit" from Python's package "scipy.optimize" over the vaccine uptake counterfactuals. We use block bootstrap to account for time dependence in the data with 1000 iterations and 30 non-overlapping blocks[50]. The 95% confidence intervals are shown and reported throughout. The confidence intervals are narrow compared to those obtained in Fig. 1. This is the case as the fitting window is longer than for the whole population (due to eligibility), and, more importantly, it includes the three phases of the logistic curve (initial growth, inflection point, and deceleration). This leads to a precise fit, with narrow confidence intervals, and which is robust to shortening the fitting window by up to April.

*Model support via synthetic control.* Figure 7a shows the estimated vaccination uptake via synthetic control for France, Germany, and Italy. Its computation is

### Table 1 Estimated parameters of the innovation diffusion model for France, Germany, and Italy.

|   | France | Germany | Italy |
|---|---|---|---|
| $p$ | 2.56E-06 | 1.98E-05 | 1.29E-06 |
|   | (2.49E-06, 2.61E-06) | (0.75E-05, 7.16E-03) | (6.11E-07, 2.67E-06) |
| $q$ | 3.01E-02 | 2.96E-02 | 3.43E-02 |
|   | (2.88E-02, 3.17E-02) | (1.58E-02, 3.10E-02) | (3.16E-02, 3.70E-02) |
| $K$ | 6.58E-01 | 6.75E-01 | 7.05E-01 |
|   | (6.40E-01, 6.83E-01) | (6.75E-01, 7.11E-01) | (6.84E-01, 7.45E-01) |
| $t_0$ | −2.65E+02 | −2.38E+02 | −2.74E+02 |
|   | (−2.74E+02, −2.53E+02) | (−2.86E+02, −5.10E+01) | (−2.91E+02, −2.41E+02) |

In brackets, the 95% confidence interval is shown.

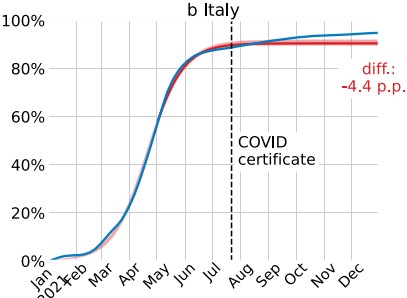

**Fig. 6 Realised and counterfactual vaccination rates for the population over age 60.** For France (**a**) and Italy (**b**), cumulative proportion of the population over age 60 who received at least one COVID-19 vaccine dose in the actual intervention deployment (blue) and in the no-intervention counterfactual scenario (red). The red shaded area is the 95% confidence interval centred around the main estimate. The counterfactual scenario is estimated via innovation diffusion theory. The black dashed vertical line is the date of the announcement of the COVID certificate.

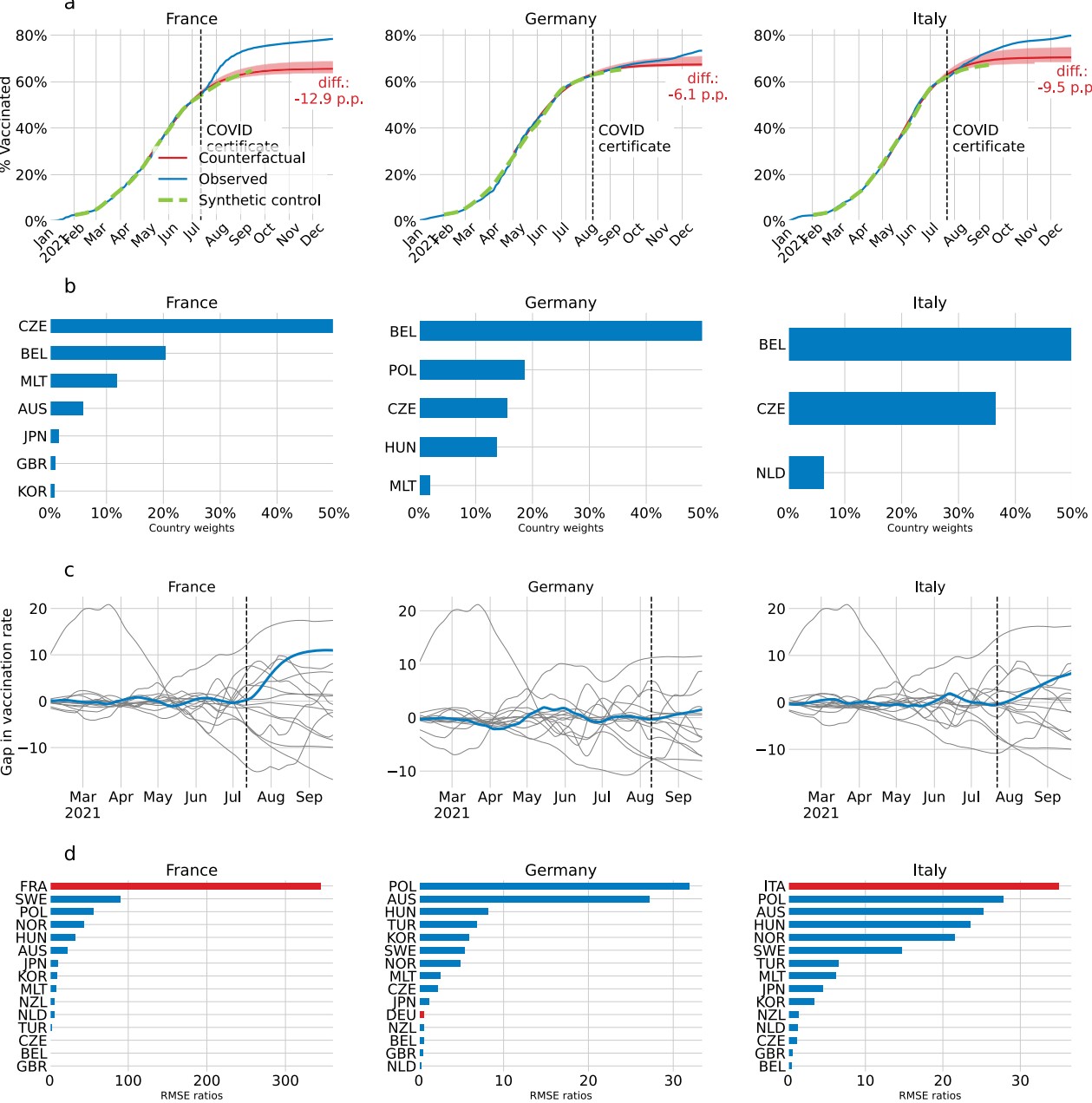

**Fig. 7 Alternative counterfactual via synthetic control. a** For France, Germany, and Italy, cumulative share of the population who received at least one COVID-19 vaccine dose in the actual intervention deployment (blue), in the no-intervention counterfactual scenario via innovation diffusion theory (red) with 95% confidence interval centred around the main estimate (shaded red), and in the no-intervention synthetic control scenario (green, dashed). The red shaded area is the 95% confidence interval of the counterfactual scenario via innovation diffusion theory, which is centred around the main estimate. Note that the synthetic control scenario ends on 22 September 2021, when the method becomes infeasible. The black dashed vertical line is the date of the announcement of the COVID certificate. **b** Country weights for synthetic control for France, Germany, and Italy. **c** Placebos for France, Germany, and Italy in blue and in grey for donor pool countries. **d** Root mean squared prediction errors (*RMSPE*) ratios for France, Germany, and Italy (red) and the donor pool countries (blue).

described below. The synthetic control for each country falls within the 95% confidence interval of the counterfactual based on innovation diffusion theory. This gives additional support for the model choice and findings.

Synthetic Control[32,51,52] provides a counterfactual based on the evolution of nontreated countries, i.e., countries that did not implement COVID certificates. This counterfactual is computed as a weighted average of the nontreated units. To this end, we define the control group as the OECD and EU countries that did not resort to COVID certificates during this period. This choice is motivated by broad socioeconomic resemblance and sufficient vaccine supply. The weights applied to the nontreated units are chosen to minimise the error of the synthetic control in the pre-treatment period. The impact of COVID certificates on vaccination is thus estimated as the difference between vaccination after the implementation of the

policy and the counterfactual – 14 countries feature in the *donor pool* of nontreated countries and include OECD and EU countries that did not implement COVID mandates before 22 September 2021 and have sufficient data availability (see list in Methods A). We posit that after this date the synthetic control method is no longer feasible due to an insufficient donor pool.

For each treated country (France, Germany, Italy), the synthetic first-dose vaccination rate ($SV_{i,t}$) is computed as a weighted average of the vaccination rates in the donor countries:

$$SV_{i,t} = \sum_{j=1}^{J} \omega_j V_{j,t}, \qquad (5)$$

where $J = 14$ is the number of countries in the donor pool, and $\nu_j$ is the weight

associated with $V_{j,t}$ and the vaccination rate in country $j$. The weights are in the interval $[0,1]$ and sum to one to avoid extrapolation[32]. They are chosen to minimise the error prior to the treatment, which occurs in $T_0$:

$$min_\omega \sum_{t=1}^{T_0}[(V_{i,t} - \sum_{j=1}^{J} \omega_j V_{j,t})^2 + \sum_{k=1}^{K} \lambda_k (X_{i,t}{}^k - \sum_{j=1}^{J} \omega_j X_{j,t}{}^k)^2]. \quad (6)$$

The weights $\omega_j$ are chosen to minimise a composite loss function that includes, on the left, the mean squared prediction error of the pre-treatment outcome, and, on the right, the mean squared prediction errors of $K$ covariates, whose respective importance is weighed by the coefficients $\lambda_k$ for $k$ between 1 and $K$. The covariates are selected on the basis of their predictive power of vaccination, and include annual GDP per capita, the average fatalities and cases over the pre-treatment period, the share of the population aged over 65, the average Mobility Index over 2020, and average vaccine acceptance over the first quarter of 2021. The covariate weights are assumed to be constant (i.e., $\lambda_k = \lambda$ for all $k$), and the weight applied to all the covariates $\lambda$ is optimised using five-fold cross-validation.

The country weights used for building the synthetic vaccination rates for France, Germany, and Italy are shown in Fig. 7b. The three synthetic vaccination rates are built as averages of the vaccination rates from European countries, except for Japan and South Korea, which each account for less than 10% of the French synthetic vaccination rate. All three heavily feature the Czech Republic and Belgium, thus meriting additional description of the policies in these countries. In Belgium, the federal government announced the adoption of COVID certificates for mass events on 19 July 2021, but the use of COVID certificates for access to day-to-day activities, such as going to cinemas, cafes, or restaurants, was not announced until mid-September, and at a regional level (Supplementary Table 3, Appendix). The situation was similar in the Czech Republic where COVID certificates were required to attend mass events from June 2021, and the announcement of their extension to day-to-day activities occurred on 21 October 2021 (Supplementary Table 3, Appendix). Further, the two control countries started requiring a valid COVID certificate for international travel by early July. The COVID certificate policies in Belgium and the Czech Republic were thus comparable to the policies in France, Germany, and Italy, up to the date of their respective announcements.

The significance of the results is assessed using placebo tests[52]. A synthetic vaccination rate $SV_{j,t}$ is built for each country $j$ in the donor pool (placebos); see Fig. 7c.

Inference is performed by comparing the ratio of the root mean squared prediction errors ($RMSPE$) of the synthetic vaccination rate for the treated country after and before the treatment to the distribution of these $RMSPE$ ratios over the placebos. The ratio between the post-intervention $RMSPE$ and pre-intervention $RMSPE$ for unit $j$ is

$$r_j = \frac{RMSPE_j(post-intervention)}{RMSPE_j(pre-intervention)}. \quad (7)$$

A $p$-value for the inferential procedure based on the permutation distribution of $r_j$ is given by the rank of the treated country's ratio divided by the size of the donor pool. The $RMSPE$ ratios and their permutation distributions for the synthetic vaccination rates in France, Germany, and Italy are shown in Fig. 7d. The causal estimates for France and Italy are statistically significant with a $p$-value equal to 0.071(=1/14), while the estimates for Germany are not significant.

Overfitting may be an issue with synthetic control. The $RMSPE$ ratios that are used for inference can be recast as follows in the absence of policy intervention:

$$\frac{RMSPE_j(out-of-sample)}{RMSPE_j(in-sample)}. \quad (8)$$

For placebos, the ratio of the post-treatment $RMSPE$ and pre-treatment $RMSPE$ is simply a ratio of the $RMSPE$ taken out-of-sample over the $RMSPE$ measured in-sample, given that the optimal weights are computed using the pre-treatment observations. This ratio is classically understood as a measure of overfitting. If the model overfits, its out-of-sample prediction error is large compared to its in-sample prediction error. As a result, overfitting synthetic control models yields spurious results whose validity is rejected by permutation tests. In the present case, overfitting is limited by using several covariates in the fit, as well as by resorting to cross-validation to select the weights attributed to these covariates.

**Impact on health outcomes**

*Vaccine effectiveness.* We estimate vaccine effectiveness against hospitalisation, ICU admission, and deaths between one week and six months after inoculation when infected with the Delta variant –the dominant strain of SARS-CoV-2 in France, Germany, and Italy over the considered time period– by taking the weighted average over the different types of vaccines (see Supplementary Table 4, Appendix). We rely on the few available studies from different contexts, thus precluding a statistical meta-study. We therefore gather conservative, lower-bound estimates below.

For mRNA vaccines (BioNTech/Pfizer and Moderna), conservative estimates for the effectiveness are 80%[33] after one dose and 93% after two doses[33–36]. The second estimate integrates the effect of waning immunity, as the protection against severe outcomes is higher than 95% up to 14 weeks after inoculation and above 90% thereafter[36]. For AstraZeneca's vaccine, comparable estimates are 90% after one dose and 85% after two doses[36]. Here, waning immunity explains the lower

effectiveness of two AstraZeneca doses versus one. Finally, for Janssen Pharmaceutica NV, the effectiveness after the single dose is estimated at 85%[37]. Overall, vaccine effectiveness appears to be similar across age groups, and we have thus opted for a common estimate[34]. The overall vaccine effectiveness against hospital admissions, ICU admissions, and deaths is approximately the same and is also similar across France, Germany, and Italy, namely 81% protection after one dose and 92% after two doses. We do not include the additional protection provided by boosters, as the calculations we perform are only concerned with individuals who were not fully vaccinated before the COVID certificate; therefore, they were not eligible for a booster shot over the period of study.

*Realised health outcomes by vaccine status.* Let $X_t$ denote the *realised* health outcome (i.e., hospital admissions and patients, ICU admissions and patients, and deaths, for a given country) at time $t$, and let $X_t(v)$ denote the same outcome by vaccine status, where $v \in \{0, 1, 2\}$ denotes the number of vaccine shots received. When the data by vaccine status are not available, we can derive them from Bayes' rule and the level of protection against the health outcome by vaccine status, $\beta(v)$, as shown above. The $X_t(v)$'s satisfy the following linear system:

$$\frac{X_t(v)}{V_{t-d}(v)} = \beta(v) \cdot \frac{X_t(0)}{V_{t-d}(0)}, \ for \ v \in \{0, 1, 2\}, \ and \ X_t(0) + X_t(1) + X_t(2) = X_t. \quad (9)$$

This system admits a unique solution, given by

$$X_t(v) = X_t \cdot \frac{\beta(v) \frac{V_{t-d}(v)}{V_{t-d}(0)}}{\sum_{v'=0}^{2} \beta(v') \frac{V_{t-d}(v')}{V_{t-d}(0)}}, \ for \ all \ v \in \{0, 1, 2\}. \quad (10)$$

Note that vaccine uptake has been lagged by $d$ days to account for the lag between infection and the health outcome, $l_{hosp}$, $l_{ICU}$, and $l_{death}$, the lag between vaccination and full effectiveness, $l_{vaccine}$, and the duration of the health hazard, $l_{stay\ hosp}$ and $l_{stay\ ICU}$, which are only relevant for hospital admissions and ICU patients. For example, for a patient who is in an ICU at time $t$, on average, their admission occurred at time $t - l_{stay\ ICU}$, their infection at time $t - l_{stay\ ICU} - l_{ICU}$, and at that time $N_{t-l_{stay\ ICU} - l_{ICU} - l_{vaccine}}(2)$, people were fully protected by vaccination.

*Counterfactual health outcomes by vaccine status.* Similarly, let $\hat{X}_t(v)$ denote the counterfactual health outcome (number of hospital admissions, ICU patients, or deaths, for a given country) at time $t$, with vaccine status $v \in \{0, 1, 2\}$. Then,

$$\hat{X}_t(v) = X_t(v) \cdot \frac{\hat{V}_{t-d}(v)}{V_{t-d}(v)} \ for \ all \ v, \quad (11)$$

where $d$ is the lag that was introduced in the previous paragraph. The estimated counterfactual number of a given health outcome at time $t$ for a given country is given by:

$$\hat{X}_t = \hat{X}_t(0) + \hat{X}_t(1) + \hat{X}_t(2). \quad (12)$$

Finally, the overall realised and counterfactual of a given health outcome, from the announcement of COVID certificates in the country until the end of 2021, are estimated respectively by

$$X_{total} = \sum_{t=T_0}^{T_1} X_t \ and \ \overline{X}_{total} = \sum_{t=T_0}^{T_1} \hat{X}_t. \quad (13)$$

The difference $\hat{X}_{total} - X_{total} = \sum_{t=T_0}^{T_1}(\hat{X}_t - X_t)$ is attributed to the adoption of COVID certificates.

*Age-stratified health outcomes.* When the data are available, we analogously compute age-stratified (i.e., 60 years old and above, and the rest of the population) health outcomes, as well as the corresponding counterfactuals. The total numbers are obtained by summing over all age groups.

*The lag parameters[53].* We assume the lag between vaccination and full effectiveness is $l_{vaccine} = 7$ days, the lag between infection and hospital admission is $l_{hosp} = 7$ days, the lag between infection and ICU admission is $l_{ICU} = 10$ days, the total number of days in ICU is 8, so that an ICU patient has been admitted $l_{stay\ ICU} = 4$ earlier, and the average lag between infection and death is $l_{death} = 14$ days. Thus, for hospital admissions, the total lag is $l_{vaccine} + l_{hosp} = 14$; for ICU patients, the total lag is $l_{vaccine} + l_{stay\ ICU} + l_{ICU} = 21$; and for deaths, the total lag is $l_{vaccine} + l_{death} = 21$.

*Confidence intervals.* For the estimation, as the vaccine effectiveness is assumed fixed, the 95% confidence intervals come only from the uncertainty of our estimation on vaccine uptake.

**Impact on the economy.** The analysis of the economic impact of COVID certificates is based on an indirect causal model: COVID certificates boost vaccination, and vaccination encourages individuals to resume economic activities, thus increasing GDP growth. The quantification exercise thus follows a two-step approach. First, we estimate the average impact of a marginal increase in vaccination rates on economic activity using two-way fixed-effect regressions based on a

large panel of data from 46 countries. Second, the estimate of the average effect of vaccination on economic activity is combined with the estimate of the uplift in vaccination obtained in section B to gauge the effect of COVID certificates on economic activity.

*Average effect of vaccine uptake on economic activity.* Our estimation is based on data from 46 countries. The identification exploits high-frequency within-country variations in vaccination rates, and assumes a static relationship whereby vaccination at time $t-28$ impacts economic activity at time $t$. We use a two-way fixed-effect regression and identify the effect through a difference-in-differences design, which assumes a common trend across countries conditional on several covariates described below (see also the descriptive statistics in Supplementary Table 5, Appendix).

The measure of economic activity used in this paper is the OECD Weekly Tracker, a proxy of weekly GDP relative to the pre-crisis trend, which is available for 46 countries with no publication delay (Methods A). It is regressed on vaccination rates along with controls as well as country and week fixed effects. To estimate the average total effect of vaccination on GDP, we use the following closed-form model:

$$T_{i,w} = \beta V_{i,w-l} + \gamma I_{i,w-l} + \eta X^f_{i,w} + \iota Z_{i,w} + \alpha_i + \delta_w + \sigma_{i,w}. \quad (14)$$

Weekly GDP is proxied by the Tracker $T_{i,w}$ and is regressed on the share of vaccinated people lagged by $l$ weeks ($l = 4$), $V_{i,w-l}$, as well as three vectors of controls, week, and country dummies. The first controls vector $I_{i,w-l}$ includes lagged cases, deaths, reproduction rate, and mobility index, which may have impacted both past vaccine uptake decisions and present weekly GDP[54]. The model also averts confounding effects that could emerge from trade and other spillovers due to the relative synchronicity of vaccination campaigns across countries by controlling for vaccination and deaths in the main trading partners ($X^f_{i,w}$). The vector $X^f_{i,w}$ is the weighted average of vaccination rates and deaths in

country $i$'s main 10 trading partners, i.e.,

$$V^f_{i,w} = \sum_{j=1}^{10} \gamma_{i,j} V_{j,w}, \quad (15)$$

where $V_{j,w}$ is the vaccination rate in trading partner $j$ and $\gamma_{i,j}$ is the share of exports from country $i$ to trade partner $j$ in total exports from country $i$. The same formula is used to build the vector of weighted average death rates in trading partners. Last, the model includes the vector of average weekly temperatures $Z_{i,w}$, which can influence virus transmission[42].

The model is estimated using data from 46 OECD and G20 countries (see Table 2). Denote by $*p < 0.1$, $**p < 0.05$, and $***p < 0.01$. The average effect of a 1 p.p. increase in the share of vaccinated people after a month is 0.052*** p.p. in weekly GDP. This order of magnitude seems plausible and implies, if the impact was permanent, that 100% vaccination uptake would increase GDP by 5.2 p.p., which broadly corresponds to 85% of the average GDP loss suffered in 2020 by the countries in the sample. This is consistent with the notion that a complete vaccination would not be sufficient to a return to pre-crisis trends due to partial vaccine effectiveness and the waning-out of vaccine-provided immunity. Adding controls for deaths and vaccination in trade partners decreases the main estimate from 0.054*** to 0.052*** by partialling out the confounding effect of trade spillovers. Finally, the third column models the direct effect by controlling for current cases, deaths, and reproduction rates. This indicates that 83% of the total economic effect of vaccination is through the direct effect on individual behaviour, while the remaining 17% is related to the effect through the impact on virus circulation. Note, however, that we do not estimate the indirect effects independently, as we do not estimate a policy response function.

Estimating the impact of vaccination on economic activity needs to take into account the uncertainty relative to the measure of the latter, proxied by the OECD Weekly Tracker. The OECD Weekly Tracker is an algorithm prediction and includes an error term that can impact the accuracy of the estimation of the parameter β in the regression above. Its standard deviation thus needs to be adjusted to account for the uncertainty from both the econometric estimation and

**Table 2 Regression results for vaccination-GDP elasticity.**

| | Baseline | Controls (trade partners) | Direct effect |
|---|---|---|---|
| Vaccinated people (per 100) | 0.054*** | 0.052*** | 0.043*** |
| | (0.044, 0.064) | (0.042, 0.061) | (0.033, 0.052) |
| Cases (lag) | 0.001** | 0.001*** | 0.002*** |
| | (−0.000, 0.001) | (0.000, 0.002) | (0.001, 0.003) |
| Deaths (lag) | −0.068*** | −0.041*** | −0.049*** |
| | (−0.103, −0.033) | (−0.075, −0.007) | (−0.086, −0.012) |
| Reproduction rate (lag) | −1.606*** | −1.305*** | −0.883*** |
| | (−1.850, −1.363) | (−1.540, −1.071) | (−1.123, −0.642) |
| Mobility Index (lag) | 0.057*** | 0.048*** | 0.054*** |
| | (0.046, 0.068) | (0.038, 0.059) | (0.044, 0.064) |
| Stringency Index (lag) | 0.014*** | 0.013*** | 0.007* |
| | (0.006, 0.023) | (0.005, 0.021) | (0.001, 0.015) |
| Temperature | 0.052*** | 0.008 | −0.014 |
| | (0.030, 0.071) | (−0.012, 0.028) | (−0.034, 0.006) |
| Vaccination of trade partners | | −0.093*** | −0.090*** |
| | | (−0.115, −0.071) | (−0.112, −0.069) |
| Deaths in trade partners | | −0.461*** | −0.453*** |
| | | (−0.595, −0.328) | (−0.586, −0.320) |
| Cases in partners | | 0.001 | 0.003*** |
| | | (−0.001, 0.003) | (0.001, 0.005) |
| GDP of partners | | 0.861*** | 0.732*** |
| | | (0.755, 0.967) | (0.628, 0.836) |
| Cases | | | −0.002*** |
| | | | (−0.002, −0.001) |
| Deaths | | | −0.086*** |
| | | | (−0.124, −0.047) |
| Reproduction rate | | | −1.667*** |
| | | | (−1.948, −1.386) |
| Country dummies | Yes | Yes | Yes |
| Week dummies | Yes | Yes | Yes |
| Observations | 4204 | 4204 | 4204 |
| R2 | 0.774 | 0.794 | 0.807 |
| Adjusted R2 | 0.766 | 0.787 | 0.800 |
| Residual Std. Error | 2.497 (df = 4059) | 2.384 (df = 4055) | 2.309 (df = 4052) |
| F Statistic | 96.581*** | 105.695*** | 112.311*** |
| | (df = 144; 4059) | (df = 148; 4055) | (df = 151; 4052) |

*p < 0.1, **p < 0.05, and ***p < 0.01

the weekly GDP measure. This can be achieved by using the multiple estimates from the OECD Weekly Tracker. The OECD provides 300 bootstrap series for the Tracker. Similar to the context of multiple imputation, we can apply Rubin's law[55] to infer the variance of $\beta$ from the estimates $\beta_i$ obtained from each of the 300 replicates and using the following formula:

$$Var(\beta) = W_M + (1 + M^{-1})B_M, \qquad (16)$$

where $M$ is the number of replicate series ($M = 300$ in our case), and $W_M$ and $B_M$ are respectively the within and between components, given by $W_M = M^{-1}\sum_{i=1}^{M}\widehat{Var}_i(\beta_i)$ and $B_M = (M - 1)^{-1}\sum_{i=1}^{M}[\beta_i - \beta]^2$. The total empirical standard deviation of $\beta$ is 0.009, which results in 95% confidence intervals of [0.034, 0.070], which are around twice the size of the confidence intervals computed in Table 1.

Results in Table 2 are to be understood as average effects; the regression analysis does not aim at capturing possibly heterogeneous effects across countries or time. The effect of vaccination on economic activity may differ across countries and regions, based on cross-country or cross-region differences in sectoral composition and institutional settings. Moreover, the effect of vaccination may also differ over time. This is the case as the protection granted by the vaccine may have a smaller impact when virus circulation is near zero. The present analysis simply aims at gauging the average impact of vaccination on the economy. Some of the sources of heterogeneity are uncovered in the robustness checks performed below.

*Economic impact of COVID certificates.* The economic impact of COVID certificates is assessed through the indirect causal model whereby COVID certificates spur vaccination, and which in turn increases economic activity. As a result, the counterfactual for economic activity (i.e., in the absence of COVID certificates) is obtained by plugging the counterfactual for vaccination rates in the regression model. The latter provides an estimate for weekly GDP absent COVID certificates by subtracting the estimation of the impact of COVID certificates on weekly GDP, denoted by $\delta_{i,w}$, from the observed weekly GDP:

$$\hat{T}_{i,w} = T_{i,w} - \delta_{i,w} \qquad (17)$$

$$\text{with } \delta_{i,w} = \beta\left(V_{i,w-l} - \hat{V}_{i,w-l}\right), \qquad (18)$$

where $\hat{T}_{i,w}$ is the counterfactual tracker and $\hat{V}_{i,w-l}$ is the counterfactual vaccine uptake.

Confidence intervals for counterfactual weekly GDP are derived from the fact that the estimate of the causal impact of COVID certificates on weekly GDP is the product of two random variables:

$$Var\left(\delta_{i,w}\right) = Var(\beta) \cdot Var\left(\hat{V}_{i,w-l}\right) + Var(\beta) \cdot E^2\left(V_{i,w-l} - \hat{V}_{i,w-l}\right)$$
$$+ Var\left(\hat{V}_{i,w-l}\right) \cdot E^2(\beta). \qquad (19)$$

$Var\left(\hat{V}_{i,w-l}\right)$ is estimated from 1,000 bootstrap runs of the logistic model, and $Var(\beta)$ is given by Rubin's law. Then, supposing that $\delta_{i,w}$ follows a normal distribution, its 95% confidence intervals are $[\delta_{i,w} \pm 1.96 * \sigma_{\delta_{i,w}}]$.

*Robustness checks.* We complemented our analysis with robustness checks regarding the statistical method, the choice of lag, the modelling assumption that the vaccine-GDP relationship did not vary substantially across the considered time period, and the choice of the measure of economic activity.

Two-way fixed effects regressions: A recent literature[56–58] has shed light on the limitations of two-way fixed effects regressions when the treatment effect is heterogeneous. Alternative estimators have been proposed[56], which limit the risk of bias by restricting the comparisons between units and times. This literature is still young, and there are currently no satisfactory options for cases where the treatment is dynamic, with a fuzzy design and in the absence of stayers or quasi-stayers. More specifically, the fuzzy design (continuous treatment, with treatment intensities that vary both in time and across units) precludes the use of the estimators introduced

by Callaway and Sant'Anna[59] or Sun and Abraham[60]. In the absence of quasi-stayers, it is impossible to estimate the time fixed effects based on the imputation estimator introduced by Borusyak et al.[61].

We assume that the effect is static, which means that the past treatments do not impact the outcome variable. The reason for this is mostly that the treatment is the cumulative vaccination rate. This implies a nested causal relationship $V_{j,t-n} \rightarrow V_{j,t-n+1} \rightarrow \ldots \rightarrow V_{j,t}$ so that it is unnecessary to model dynamic effects. Under the alternative assumption that the effect is dynamic, the $DID_l$ estimator[62] could be applied but our experiments were inconclusive. This seems to result from the fact that identification is based on the timing of the intervention rather than the intensity of the treatment; yet, in our paper, we focus on the intensity. Further, their $DID_l$ estimator is a weighted average of DID between first-time switchers in $t - l$ and not-yet-switchers in $l$. As a result, the estimation for each possible value of $l$ is based on a small number of observations (e.g., 44 for $l = 1$), and the results are not credible.

To nevertheless validate our findings with a different statistical method, we reproduced the regressions in Table 2 without week dummies. Note that this model is underspecified, so the results can only be seen as indicative. For the preferred specification, estimates of the vaccine effect on GDP are of the same order of magnitude, albeit substantially smaller (0.034***). The model, which includes week dummies, remains more plausible, as it seems critical to control for the very large shocks caused by successive COVID infection waves across the globe.

Choice of lag: The main model regresses weekly activity over the vaccination rate lagged by $l$ weeks, which is equal to 4 in the favourite specification. The coefficients of interest were computed for all values of $l$ between 1 and 10. Figure 8a shows that the main estimates are robust to the choice of lag parameter.

Time-varying relationship between vaccination and GDP: The regression analysis used to analyse the economic impact of vaccination yields an average effect which is assumed to be a valid approximation of the economic impact of the increment in vaccination caused by the COVID certificates. To test the robustness of this hypothesis, we use causal machine learning to produce a time-varying estimate of the economic effect of the vaccine. We considered a model with time-varying effects using Double Machine Learning[63,64]. This approach uses machine learning to capture non-linearities and complex interactions while correcting for the bias caused by penalised loss functions. More specifically, we used the R-learner[64], which allows us to include a large number of interaction terms while averting overfitting by applying a penalty term such as the $L^2$-norm. We apply the R-learner to the regression of weekly GDP on vaccine uptake by including as interaction terms the complete set of overlapping period dummies, $\{P_t^w = I_{(t \leq w)}\}_{w \in [1,M]}$, i.e., the dummy $P^w$ is equal to 1 at times $t$ prior to week $w$, and zero otherwise. Using overlapping period dummies rather than week dummies in the context of a penalised regression allows for smoother time-varying estimates, and both the value of the coefficient associated with the period dummy and the difference between two consecutive coefficients are penalised[65]. This approach yields a time-varying coefficient $\beta$ (Fig. 8b). Given that the interaction terms are common to all countries, this model estimates a time-varying effect without allowing for cross-country heterogeneity. The estimated impact lies between 0.035 and 0.060, with a 95% confidence interval of approximately 0.02 percentage points on average. We note that the estimates do not vary significantly across time, and our average estimator of 0.052 is consistent with this method.

Measure of economic activity: The main analysis of the impact of vaccination on economic activity is based on the OECD Weekly Tracker, which proxies weekly GDP relative to the pre-crisis trend. Additional regression analyses were performed to ensure that the results are robust to the choice of an economic activity metric. Supplementary Table 6 (Appendix) shows estimation results of regressions on both the Mobility Index and official quarterly GDP. The former is a high-frequency proxy for economic activity, although its relationship with GDP is not straightforward, and the latter is the official low-frequency measure of economic production. The regression on the Google Maps mobility index yields results that are comparable to the main regression (Table 2), although with a smaller direct effect of vaccination. From our preferred specification, an increase of 10 p.p. in vaccination increases mobility by 6.9 p.p. This effect is one third larger than the effect on

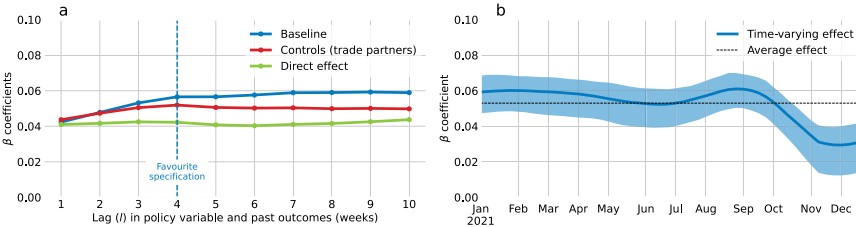

**Fig. 8 Robustness checks for varying lags and time-varying impact of vaccination. a** Main regression coefficients for various lag values. **b** Time-varying estimate of the impact of vaccination on economic activity. The blue shaded area is the 95% confidence interval obtained from 100 bootstrap samples and centred around the main estimate (blue). The black dashed line indicates the average effect across time.

the Weekly Tracker, which seems consistent with the fact that mobility was only partially correlated with economic activity as by the start of 2021 households and firms had both adapted to the pandemic constraints (through working from home, e-commerce, etc.). As a result, around two thirds of the increase in mobility triggered by vaccination translates into an increase in economic activity. The regression on quarterly GDP year-on-two-year growth rates cannot include regressors lagged by 4 weeks, which implies that the baseline model cannot be estimated. Note that, the use of year-on-two-year growth rates avoids the strong base effects in year-on-year (resp. quarter-on-quarter) series around one year (resp. one quarter) after the first lockdowns. The direct effect model yields an estimate of the impact of vaccination (0.028) on GDP that is close to the one that was obtained from the regression on the Weekly Tracker (0.043), although it is much less precise given that the identification cannot rely on the high-frequency variations in both vaccination and economic activity. The results obtained from regressions on these two alternative measures confirm the robustness of the main effect captured with greater precision from the regression on the OECD Weekly Tracker.

**Reporting summary**. Further information on research design is available in the Nature Research Reporting Summary linked to this article.

## Data availability
All data used in the study is accessible from publicly available sources as detailed in Methods A. In addition, the used data is available on https://barypradelski.com/projects/covid-certificates/.

## Code availability
The code is available on https://barypradelski.com/projects/covid-certificates/.

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

## Acknowledgements
We are grateful for comments and suggestions by Laurence Boone, Anne Bucher, Maria Demertzis, Mathias Dewatripont, Luiz de Mello, Thierry Pech, Alain de Serres, David Turner, and seminar and expert consultation participants at Bruegel (Brussels), French Scientific Council (Paris), European Centre for Disease Prevention and Control (Stockholm), Institute for Interdisciplinary Innovation in healthcare (I3h, Brussels), Institut Pasteur (Paris), OECD (Paris), and Terra Nova (Paris). This work has been partially supported by Bruegel, French National Research Agency (ANR) through grant ANR-19-CE48-0018-01, and University Paris Dauphine - PSL. All errors are ours.

## Author contributions
M.O.-B, B.S.R.P., and N.W. designed the study, built the model, collected data, finalised the analysis, interpreted the findings, and drafted the manuscript. L.G.-J. collected and analysed data. P.A.g, P.Ar., A.F., P.M., and G.B.W. provided guidance, interpreted the findings, and commented on and revised previous versions of this manuscript. All authors read and approved the final manuscript. The corresponding authors had full access to all the data used in the study and had final responsibility for the decision to submit for publication.

## Competing interests
A.F. is a member of the French COVID-19 Scientific Council and a member of the French COVID-19 Vaccine Strategy Committee. PM chairs the French Council of Economic Analysis, an independent council attached to the Prime Minister. G.B.W. is a member of the G20 High Level Independent Panel on Financing the Global Commons for Pandemic Preparedness and Response. The authors declare no other competing interests.
