## [Peer Review File · Nature Communications]

The effect of COVID certificates on vaccine uptake, health outcomes, and the economyREVIEWER COMMENTS

Reviewer #1 (Remarks to the Author):

Dear authors,
thanks for the opportunity to review this valuable piece of work.
My comments are included in the attached file. I hope you will find them useful.
All the best.

Reviewer #2 (Remarks to the Author):

TO THE AUTHORS

Despite the recent shift of public attention away from COVID, the submitted paper's topic remains important and with future significance. The authors employ statistical and computational methods to estimate the impact of the COVID-19 mandates/certificates on vaccine uptake and extrapolate these results to evaluate the policies' impact on health outcomes and GDP. I view the paper as a useful contribution and complement to previous work on the same topic, specifically two existing prior studies: Mills and Ruttenauer (2021), hereafter MR and Karaivanov, Kim, Lu and Shigeoka (2021), hereafter KKLS, using (in part) different data sources and methodologies. The study is carefully and rigorously done and the results are in line (or larger in magnitude) from those in existing work, although the relevant literature and the paper's innovation and contribution to it is not sufficiently well-explained. My main concerns lie with the innovation diffusion model that the authors use to estimate the effect on vaccine uptake and the extrapolation of the results to health and economic outcomes. Detailed comments follow.

Major comments

1a. I wonder whether the innovation diffusion model is the best way to think of COVID-19 vaccine uptake, especially in the early months. Vaccine eligibility was heavily regulated and proceeded in stages, based on age and other factors in most countries. The role of vaccine supply and eligibility should be clarified and explained, e.g., it is hard to believe the assumption of the model that adoption in January-April 2021 (even among 60+ adults) was purely based on a "coefficient of innovation" and "coefficient of imitation". How are the eligibility and supply constraints incorporated? At best the model parameters can be mechanically re-interpreted (the authors seem to fit from May onwards in the main results, please specify the start date) to proxy such constraints but this needs to be clearly stated (and its usefulness justified). In addition, discussion on the latter period, e.g., from June or July onwards when eligibility and supply constraints were essentially gone for first doses should be done in the context of the model (e.g., possibly use $K(t)$ instead of K on p.11 and in the estimation).

1b. Related to point 1, on p.12 - the estimation for the 60 and older group begins in January and seems to assume all were eligible at the same time, is that correct? If not, how does it affect the results?

2. Technical questions related to the model

- shouldn't p , the coefficient of innovation depend on risk-assessment in the context of COVID (e.g., current cases, deaths, etc.)?

- what assumptions are being made about potential heterogeneity across people?

- it is said on p.11 that the model assumes absence of major shocks but the Delta variant spread in the period of study which affects risk and hence potentially vaccine uptake. Need to discuss.

3. If issues 1-2 cannot be addressed in a satisfactory way, the authors may consider making the synthetic control robustness check their main estimation method, however, that will diminish the contribution of the paper since MR use the same method.

4. When addressing health outcomes, e.g. averted ICU admissions on p.5, what other behavioural factors may exist that may offset the identified effect, e.g., higher uptake may mean engaging in

more social contact or there may be selection effects? These factors can significantly affect the counterfactual ICU numbers estimated.

5. How reliable are the "OECD weekly tracker" numbers based on "google trends search results and machine learning"? Are there any studies that have looked at that ex-post, relative to official national statistics? In particular, I assume these numbers come with standard errors or confidence bands but it appears that this uncertainty is not incorporated in the calculation of confidence intervals for the effect of increased vaccine uptake on GDP.

6. Do the estimates on vaccine effectiveness in Table 3 include medical evidence from the Delta variant? Given that the analysis ends in December 2021 I believe that this is important and hence a wider intervals for these estimates may need to be considered. More specifically, are the numbers listed on top of p.16 the ones used in computing the counterfactuals, the $\beta(v)$? In any case, it is warranted to include confidence intervals around these numbers. In general, how the CIs of the estimates are computed, what is included and what is not, should be made clearer.

7. The GDP estimates appear to use standard OLS TWFE regressions which have been criticized in the recent methodological literature (as the authors correctly acknowledge on p.18). Methods and estimation routines for dynamic staggered interventions do exist now, e.g., in Stata there is `csdid` by Callaway and San'tAnna or `eventstudyinteract` by Sun and Abraham among other routines. This part of the paper would be improved by using an appropriate estimation method robust to heterogeneity and dynamic treatment effects across cohorts.

Minor comments

1. p.2, top, the authors say: "The impact of COVID certificates on vaccine uptake, health outcomes, and the economy has not yet been investigated." This is (partially) incorrect - the authors need to engage more carefully the existing literature on the topic, specifically the MR and KKLS papers which have been publicly available since at least October 2021 and potentially other works that have appeared in the meantime.

2. p.5, the calculation of the impact on GDP appear to assume a linear effect; this needs to be clarified/justified.

3. p.11, what is $f(t)$? Is it the same as $x(t)$?

4. I couldn't find in the submitted materials the estimates and standard errors for the diffusion model parameters. Also, no detailed description of the estimation method was provided (is it simulated method of moments?) Please add details.

5. p.11 - what are the exact starting dates for the fitting exercise? For instance, if starting in April issues with supply and eligibility are important. Need to discuss how this may bias the estimates. At the very least robustness checks with later initial dates (e.g. June onwards) should be performed.

6. p.12, the calculation of counterfactual second doses seems to assume a 3-week gap. What is the actual average gap in the data? The assumed gap affects the health outcome estimates (e.g., may over-estimate second doses), need to check robustness.

7. p.17 - the use of the reproduction rate in the controls - how is it computed? It usually uses cases/deaths itself plus policy/behaviour, why include separately? Discuss. Also, if already control for cases, deaths etc. directly what would be any residual effect from temperature? Please explain.

8. Why are the confidence intervals so narrow in Fig 7, compared to Fig. 1?

*References

Mills, M. and T. Ruttenauer (2021), "The effect of mandatory COVID-19 certificates on vaccine

uptake: synthetic-control modelling of six countries", *Lancet Public Health* 7(1):E15-E22.

Karaivanov A., D. Kim, S. Lu, H. Shigeoka (2021), "COVID-19 Vaccination Mandates and Vaccine Uptake", preprint, <https://www.medrxiv.org/content/10.1101/2021.10.21.21265355v3>

Reviewer #3 (Remarks to the Author):

NCOMMS-22-02909: "The effect of COVID certificates on vaccine uptake, health outcomes, and the economy"

This manuscript investigates the effect of mandatory vaccination certificates on vaccine uptake, health outcomes and the economy based on data from France, Germany and Italy. It uses innovation diffusion modelling to construct counterfactual outcomes. The results reveal a consistent impact of certificates on vaccine uptake, and a beneficial effect on health outcomes in France and Italy.

The manuscript provides an important contribution to the literature. The analysis on vaccine uptake seems straightforward, also given the additional robustness checks. However, I have several comments on the other outcomes.

1. Significance

This study is innovative, and it adds important findings to the literature on vaccine certificates. However, the statement "the incentive effect of COVID certificates on vaccine uptake, health outcomes, and the economy has not yet been investigated" (abstract and p.2, l.40) is not true. For instance (and as mentioned later on), Mills & Rüttenauer (2022) investigate the effect of certificates on vaccine uptake and on case rates in the supplement. The paper needs to clarify its additional contribution.

Also, the working paper by Karaivanov et al. (2021) seems relevant.

2. Validity of results / interpretation

It should be clarified upfront that the analyses on additional outcomes (deaths, ICU admission, GDP) are not independent but are based on the results on vaccination uptake. Or do I misinterpret something here?

It is stated several times that the paper finds robust effects of certification on health outcomes (e.g. p. 6, l. 216). However, the results on health outcomes in Germany are not statistically significant. The conclusions should clarify that (also in the abstract).

I am concerned that the results on health outcomes might be confounded by the onset of omicron. The results (Figure 2) only indicate substantial differences between observed and the estimated counterfactual in December. However, according to Our World in Data, around one third of cases was already omicron in December (and I guess this an under-estimation). This might drive the lower deaths and ICU rates in December, and it is difficult to say how much of it is attributable to the vaccination rate (Wolter et al. 2022).

3. Data

Subsection A (p. 8) mentions data on vaccine acceptance. However, I could not find information where this is used in the analyses?

Can you please give some information on the accuracy of the OECD weekly tracker?

Could you clarify why the paper only looks at first doses in the main analysis? Could there be an additional effect on second doses?

4. Analytical approach

How does the statistical approach account for the fact that the countries have adopted other non-

pharmaceutical measures, such as closing public venues and distancing rules, during the autumn of 2021? How do we know the differences in health outcomes between actual and estimated counterfactual health outcomes are due to the certificates and not due to the other measures? I see there is a stringency index included in the analysis on economic outcomes, but this seems similarly relevant for the analysis on health outcomes.

Similarly, how might the booster rollout in November and December affect the results on health outcomes? Again, it coincides with the period of the largest "treatment effect".

There is no information on how confidence intervals for Figure 2 / method C are calculated, or I am missing it. How do we account for the fact that the effect of vaccination on health outcomes as well as the counterfactual vaccination rates are estimates – both associated with insecurity?

As far as I understand, the analysis of GDP (Figure 3) uses a single coefficient (beta_vaccinated from Table 4?) to estimate the counterfactual. This assumes that we have a homogeneous effect of vaccines on health outcomes. However, wouldn't we expect that this effect changes over time (e.g. declining marginal benefit with increasing overall vaccination rate, different effects because of omicron)? – Is there a way of making these estimates more flexible?

5. Clarity and context

The manuscript is clear and nice to read, sufficient context is provided. I have just two minor suggestions.

Given the similarity to Mills & Rüttenauer (2022), also in terms of countries, it might be helpful for the reader to know how results compare to each other.

On page 1, the paper is placed in the context of vaccine hesitancy and refusal. Do we know which groups are most responsive to certification? Is it those who oppose vaccines, or rather young people who feel no health-related needs to get vaccinated, or those who are previously undecided for instance?

References

Mills, M. C., & Rüttenauer, T. (2022). The Effect of Mandatory COVID-19 Certificates on Vaccine Uptake: Synthetic-Control Modelling of Six Countries. *The Lancet Public Health*, 7(1), e15-e22. [https://doi.org/10.1016/S2468-2667\(21\)00273-5](https://doi.org/10.1016/S2468-2667(21)00273-5)

Karaivanov, A., Kim, D., Lu, S. E., & Shigeoka, H. (2021). COVID-19 Vaccination Mandates and Vaccine Uptake (Vol. 597). <https://doi.org/10.1101/2021.10.21.21265355>

Wolter, N., Jassat, W., Walaza, S., Welch, R., Moultrie, H., Groome, M., Amoako, D. G., Everatt, J., Bhiman, J. N., Scheepers, C., Tebeila, N., Chiwandire, N., Du Plessis, M., Govender, N., Ismail, A., Glass, A., Mlisana, K., Stevens, W., Treurnicht, F. K., . . . Cohen, C. (2022). Early assessment of the clinical severity of the SARS-CoV-2 omicron variant in South Africa: a data linkage study. *The Lancet*, 399(10323), 437–446. [https://doi.org/10.1016/S0140-6736\(22\)00017-4](https://doi.org/10.1016/S0140-6736(22)00017-4)

Review of “The effect of COVID certificates on vaccine uptake, health outcomes, and the economy”

12/02/2022

Thanks for the opportunity to review this is an insightful piece of work exploring the effect of COVID certificates on vaccine uptake, health outcomes in France, Italy, and Germany. The authors evaluate effects of COVID certificates on vaccine uptake, as well as the effect of increases in vaccine uptake on health outcomes. The authors also measure the effect of increased vaccine uptake on economic activity using the weekly OECD GDP tracker, and estimating a two-way fixed effect specification. The results suggest that COVID certificates substantially increase vaccine uptake after the introduction, prevented many hospitalizations and deaths, and improved the economic performance. These results add to a very scarce body of literature evaluating the effects of public health measures introduced to curb the negative effects of the COVID-19 pandemic. As such, this paper is original and potentially of great value.

However, I do have some reservations, mostly about the data and empirical methods employed.

1. Effect of COVID certificates on vaccine uptake.

The authors use high-frequency country-level aggregate data and propose a non-conventional approach to obtain counterfactual levels for vaccination levels in absence of COVID certificates based on innovation diffusion dynamics. The approach is convincing and interesting. The authors validate these results using a synthetic control approach. I have a few comments on this part of the analysis:

- a) The authors fail to discuss how similar/different the COVID certificate policies are between the three main countries analysed.
- b) Similarly, although the authors mention that “the three countries also have comparable per-capita vaccine supply, demographics, health infrastructure, and economies”, they fail to provide substantiative support to these statements. To this end, a table with descriptive statistics would help.
- c) France, Italy and Germany introduced COVID certificates within a month (roughly), with France moving first. Could there be an anticipation effect in Italy and Germany biasing the results obtained for these two countries? What would be the direction of this bias? This is never discussed in the manuscript and I encourage the authors to engage with this potential problem.
- d) In section Methods B, the authors do not mention explicitly what is the unit of time used. Day? Week?
- e) The authors mention that parameters of the innovation diffusion model are estimated with a least-squares approach, both for the main results and for the age-stratified analyses for France and Italy. However, the paper does not report neither the results of these regressions, nor any measure of goodness of fit (if appropriate). This should be included for transparency.
- f) The similarity between synthetic control and counterfactual based on innovation diffusion is mentioned as a reassuring sign. Given the great heterogeneity in content and timing of the COVID policies introduced in donor pool countries, and the geographic and cultural closeness of some countries to some of the countries of interest (e.g. Belgium vs. France and Germany), why should the reader be reassured by these results? This is not immediately clear to me, and would probably deserve some more discussion
- g) Similarly to point d), the synthetic control method puts in all three cases a 50% weight on one single donor. I believe the authors could partially address point d) discussing how the major donors for the different countries of interests compare in terms of (lack of) policies and timing of introduction of COVID certificate, to convince the reader that they act as legitimate controls.

2. Impact of COVID certificates on health outcomes

The authors declare that they “estimate the impact of vaccine uptake on health outcomes” and that they “construct counterfactuals for second-dose vaccine uptake based on the observed lag between the first and second doses”. They also declare to “focus on the direct protection provided by vaccines, but omit the contribution of vaccines to reducing overall transmission and the fact that COVID certificates may alter epidemic dynamics”. My assessment is that this section presents some problems, and would benefit from some changes.

- a) The authors should make clear that their approach is unable to control for changes in population behaviour, natural immunity, and changes in disease severity. I believe this represents a major limitation of this study, and above all limits the author’s ability to interpret their results as causal estimates. I believe that some interpretations need to be toned down.
- b) The paragraph on page 3 where the authors briefly introduce the way the counterfactual is constructed for this analysis is not clear, especially where the authors state that “we construct counterfactuals for second-dose vaccine uptake based on the observed lag between the first and second doses”. Although there is specific Methods section, I suggest to rephrase in order to make clearer in simple terms how you then obtain the estimates listed in the “Findings” paragraph.
- c) In section Methods C, it would help if the authors were more explicit what vaccine statuses $v \in \{0,1,2\}$ represents. I assume it is the number of vaccine shots, but this does not come across from the text currently.
- d) Although I understand that you are likely limited by data availability, I strongly encourage the authors to explore any potential opportunity to find data at a higher level of granularity (e.g. regional, local authorities) if no individual data is available. Whilst an aggregate analysis of vaccine uptake response to COVID certificates can be justified and meaningful, the impact of vaccines on health outcomes cannot be studied appropriately based on population level indicators at such a high level of aggregation and using a weighted average effectiveness rates of vaccines obtained from the mix of vaccines administered in the different countries.

3. Impact on the economy

The authors estimate the effect of increases in vaccination rates on economic activity using a two-way fixed effect (TWFE) regression. This empirical approach seems justified, although the authors acknowledge a growing recent literature to address the limitations of TWFE. I have four main comments about this part of the manuscript:

- a) The OECD weekly tracker seems like a great source of data. However, the effect relies on a debatable counterfactual based on a projected pre-pandemic growth trend. To ensure that the results are robust to this quite strong and crucial assumption, the authors should at least attempt an analysis – even at lower frequencies (e.g. monthly) – with a different indicator of economic activity. Alternatively, the result regarding the impact on economic activity should be clearly framed as a measurement of a very specific outcome, that is the OECD weekly tracker.
- b) Similarly to the section on health outcomes, this level of granularity in the data fails to capture a potentially great heterogeneity in the effects across regions/areas with different economic/industrial structures. This should be better acknowledged in the paper.
- c) The authors should make an effort in discussing the identifying assumptions underpinning their TWFE approach. Additionally, I believe that a proper discussion of the following elements is missing: how control and treatment countries are defined, what the actual treatment is, how timing of COVID certificate policies relates to differential treatment timing, and how all this could affect estimation results.
- d) Related to point c), given the focus of the author’s on three countries (France, Italy, Germany), the use of a TWFE with no specific treatment/control may not be the best approach. Did the author consider – as in the first part of the work – estimating the economic

impact using a synthetic control approach, looking for proper comparators for each country of interest?

- e) In section Methods D (page 18, below Table 4), I encourage the authors to make the link between the counterfactual vaccine coverage obtained with in the section on “Effect of COVID certificates on vaccine uptake” more explicit. The authors introduce their way to construct the counterfactual with an equation, but repeating this link in words would facilitate the reader.
- f) In section Methods D or in an appendix, the manuscript would benefit of a table showing descriptive statistics for the variables used in the regressions in Table 4.

Overall, the study provides a great overview of the potential effects of these policies. However, building on the previous lines, I would argue that the identification strategies used to study the effect on health outcomes and economic activity are not sufficiently strong to make causal claims. This does not diminish the value of the contribution, but calls for some rewording throughout the manuscript.

Again building on my discussion above, the interpretation of the milder results for Germany could also be the result of a bias due to anticipation of stricter measures, since this country was the last of the three analysed to introduce COVID certificates.

The article makes an appropriate use of references. The only suggestion that I can make is to engage more with the (albeit limited) macroeconomic literature on the effects of Covid shocks to support the discussion about potential mechanisms for the effect measured on economic activity.

The abstract should be edited depending on changes to the rest of the manuscript, but is otherwise well written and self-contained. My only comment is that referencing in the abstract is likely not appropriate/necessary.

I hope I did not misinterpret anything, and do apologize if that is the case. Thanks again for this valuable piece of work.

Dear reviewer,

Thank you for appreciating our work and for your insightful comments and detailed suggestions to improve our article. We tried to address all your points detailed in the table below and have changed the manuscript accordingly.

We believe that the manuscript has improved and hope that you like the revised version.

Best wishes

The authors

NB. In our responses below, additional text is marked **bold** in case it is an addendum to an existing sentence that is stated in full for ease of readability.

Reviewer #1:	Response
Thanks for the opportunity to review this is an insightful piece of work exploring the effect of COVID certificates on vaccine uptake, health outcomes in France, Italy, and Germany. The authors evaluate effects of COVID certificates on vaccine uptake, as well as the effect of increases in vaccine uptake on health outcomes. The authors also measure the effect of increased vaccine uptake on economic activity using the weekly OECD GDP tracker, and estimating a two-way fixed effect specification. The results suggest that COVID certificates substantially increase vaccine uptake after the introduction, prevented many hospitalizations and deaths, and improved the economic performance. These results add to a very scarce body of literature evaluating the effects of public health measures introduced to curb the negative effects of the COVID-19 pandemic. As such, this paper is original and potentially of great value. However, I do have some reservations, mostly about the data and empirical methods employed.	Dear reviewer, thank you for your remarks and thoughtful comments. We have aimed to address them in their entirety and are detailing our changes in this table. We believe that the paper has substantially improved after this revision.
1. Effect of COVID certificates on vaccine uptake. The authors use high-frequency country-level aggregate data and propose a non-conventional approach to obtain counterfactual levels for vaccination levels in absence of COVID certificates based on innovation diffusion dynamics. The approach is convincing and interesting. The authors validate these results using a synthetic control approach. I have a few comments on this part of the analysis:	—————
a) The authors fail to discuss how similar/different the COVID certificate policies are between the three main countries analysed.	Following your suggestions, we have added the following sentence in the “Introduction” section: “While COVID certificates were required throughout France and Italy, in Germany they were only required in regions where the seven-day incidence was above 35 per 100,000 (see Methods A for detailed lists of the

	regulations in each country).” Further, we added the following sentence in the “Discussion” section: “By contrast, COVID certificates were introduced more gradually in Germany with different rules and enforcement levels across its federal states, depending on the local incidence.”
b) Similarly, although the authors mention that “the three countries also have comparable per capita vaccine supply, demographics, health infrastructure, and economies”, they fail to provide substantive support to these statements. To this end, a table with descriptive statistics would help.	The following indicators have been added in Table 1 Methods A (and referred to in the main text): Demographics Median age in 2015 (years): France 41.2 Germany 45.9 Italy 45.9 Source: UN Population Division, World Population Prospects, 2017 Revision, https://esa.un.org/unpd/wpp/Download/Standard/Population/ Old-age dependency ratio in 2020 (This is the ratio of the number of people older than 64 relative to the number of people in the working-age (15-64 years). Data are shown as the proportion of dependents per 100 working-age population.) France 33.69% Germany 33.70% Italy 36.57% Source: World Development Indicators - World Bank (2021.07.30), http://data.worldbank.org/data-catalog/world-development-indicators Health infrastructure. Health expenditure per person in 2019. (The sum of public and private annual health expenditure per person) France 5,493 international-\$ Germany 6,739 international-\$ Italy 3,998 international-\$ Source: WHO, Global Health Observatory (GHO), https://ghoapi.azureedge.net/api/ Percentage of population covered by health insurance (Estimate of health insurance coverage as a percentage of total population. Coverage includes affiliated members of health insurance or estimation of the population having free access to health care services provided by the State.) France 99.9% (year: 2011) Germany 100% (year: 2010) Italy 100% (year: 2010) Source: Scheil-Adlung, Xenia (2014), Universal Health Protection: Progress to Date and the Way Forward, International Labour Organization. OECD.Stat Economic indicators

	GDP per capita in 2020: France 42,026 international-\$ Germany 50,922 international-\$ Italy 38,992 international-\$ Source: World Development Indicators - World Bank (2021.07.30), https://datacatalog.worldbank.org/search/dataset/0037712 Income inequality in 2010 (Shown is the Gini – higher values indicate higher level of inequality – for equivalised household income. Market Income; Disposable income) France 0.51; 0.29 Germany 0.52; 0.29 Italy 0.50; 0.33 Source: Incomes across the Distribution Database, http://www.lisdatacenter.org
c) France, Italy and Germany introduced COVID certificates within a month (roughly), with France moving first. Could there be an anticipation effect in Italy and Germany biasing the results obtained for these two countries? What would be the direction of this bias? This is never discussed in the manuscript and I encourage the authors to engage with this potential problem.	Thank you for this comment. Indeed, possible spillover effects need to be addressed, especially regarding the German case where COVID certificates were adopted one month after France. Assessing the impact of COVID certificates from the date it was announced in the country provides a more conservative estimate, as the potential impact of other countries' announcements is neglected. As a robustness check, we have estimated the impact of COVID certificates in Germany starting at the French announcement instead and observed a slightly higher effect (i.e., 1.1 p.p. more by the end of the year)  Following your suggestion, we have added: - (In the main part) “Last, we did not find sizeable spillover effects between the announcements of Covid certificates in France, Germany, and Italy (see Methods B).” - (In Methods B) “For robustness, we estimated the impact of COVID certificates in Germany starting at the French announcement instead, and observed a slightly higher effect (i.e., 1.1 p.p. more by the end of the year). Assessing the impact of COVID certificates from the date it was announced in the country results in a more conservative estimate, as the potential impact of other countries' announcements is neglected.”

d) In section Methods B, the authors do not mention explicitly what is the unit of time used. Day? Week?	Replaced “at date t” with “on day t”.
e) The authors mention that parameters of the innovation diffusion model are estimated with a least-squares approach, both for the main results and for the age-stratified analyses for France and Italy. However, the paper does not report neither the results of these regressions, nor any measure of goodness of fit (if appropriate). This should be included for transparency.	We have now included Table 4 in Methods B with the parameters obtained in our estimates through mean-square minimisation; standard errors come from block bootstrap.
f) The similarity between synthetic control and counterfactual based on innovation diffusion is mentioned as a reassuring sign. Given the great heterogeneity in content and timing of the COVID policies introduced in donor pool countries, and the geographic and cultural closeness of some countries to some of the countries of interest (e.g. Belgium vs. France and Germany), why should the reader be reassured by these results? This is not immediately clear to me, and would probably deserve some more discussion	Thank you for pointing this out. We reworded from “validation” to “support”. Also note that this finding may not come at a surprise as the comparison is over a (relatively short) time-period. Regarding the similarity of the two models, we have reworded the respective paragraph, in particular adding: “As synthetic control is robust to shocks that are common to all countries, this suggests that around the period of the introduction of COVID certificates exogenous shocks, such as the rise of the Delta variant from late June, did not crucially pollute our estimates.”
g) Similarly to point f), the synthetic control method puts in all three cases a 50% weight on one single donor. I believe the authors could partially address point d) discussing how the major donors for the different countries of interests compare in terms of (lack of) policies and timing of introduction of COVID certificate, to convince the reader that they act as legitimate controls.	Thank you for pointing this out. We added in “Methods B”: “In Belgium, the federal government announced the adoption of COVID certificates for mass events on 19 July 2021, but the use of COVID certificates for access to day-to-day activities, such as going to cinemas, cafes, or restaurants, was not announced until mid September, and at a regional level. The situation was similar in the Czech Republic where COVID certificates were required to attend mass events from June 2021, and the announcement of their extension to day-to-day activities occurred on 21 October 2021. Further, the two control countries started requiring a valid COVID certificate for international travel by early July. The COVID certificate policies in Belgium and the Czech Republic were thus comparable to the policies in France, Germany, and Italy, up to the date of their respective announcements.” Footnotes: Belgium https://www.adalovelaceinstitute.org/project/international-monitor-vaccine-passports-covid-status-apps/#belgium-10 Czech Republic https://www.euractiv.com/section/politics/short_news/czechia-refuses-italian-way-says-no-to-obligatory-covid-pass-for-workers/ https://www.expats.cz/czech-news/article/coronavirus-update-oct-21-2021 https://www.expats.cz/czech-news/article/coronavirus-update-july-9-2021
2. Impact of COVID certificates on health outcomes The authors declare that they “estimate the impact of vaccine uptake on health outcomes” and that they “construct counterfactuals for second-dose	_____

vaccine uptake based on the observed lag between the first and second doses”. They also declare to “focus on the direct protection provided by vaccines, but omit the contribution of vaccines to reducing overall transmission and the fact that COVID certificates may alter epidemic dynamics”. My assessment is that this section presents some problems, and would benefit from some changes.	
a) The authors should make clear that their approach is unable to control for changes in population behaviour, natural immunity, and changes in disease severity. I believe this represents a major limitation of this study, and above all limits the author’s ability to interpret their results as causal estimates. I believe that some interpretations need to be toned down.	We have added the following in the section “Impact of COVID certificates on health outcomes”: “We focus on the direct protection provided by vaccines, but omit the contribution of vaccines to reducing overall transmission and the fact that COVID certificates may alter epidemic dynamics through, for example, behaviour changes or different patterns of increasing natural immunity.”
b) The paragraph on page 3 where the authors briefly introduce the way the counterfactual is constructed for this analysis is not clear, especially where the authors state that “we construct counterfactuals for second-dose vaccine uptake based on the observed lag between the first and second doses”. Although there is a specific Methods section, I suggest to rephrase in order to make clearer in simple terms how you then obtain the estimates listed in the “Findings” paragraph.	Rephrased as follows: “To estimate the impact of vaccine uptake on health outcomes, we construct counterfactuals for second-dose vaccine uptake by assuming the same ratio between second and first dose uptake in the counterfactual and realised scenarios (Methods B).”
c) In section Methods C, it would help if the authors were more explicit what vaccine statuses $v \in \{0,1,2\}$ represents. I assume it is the number of vaccine shots, but this does not come across from the text currently.	We have made the following clarification: “Let X_t denote the realised health outcome (i.e., hospital admissions or patients, ICU admissions or patients, and deaths, for a given country) at time t, and let $X_t(v)$ denote the same outcome by vaccine status, where $v \in \{0,1,2\}$ denotes the number of vaccine shots received.”
d) Although I understand that you are likely limited by data availability, I strongly encourage the authors to explore any potential opportunity to find data at a higher level of granularity (e.g. regional, local authorities) if no individual data is available. Whilst an aggregate analysis of vaccine uptake response to COVID certificates can be justified and meaningful, the impact of vaccines on health outcomes cannot be studied appropriately based on population level indicators at such a high level of aggregation and using a weighted average effectiveness rates of vaccines obtained from the mix of vaccines administered in the different countries.	We agree that a finer level of granularity may overcome some averaging effects. For health outcomes this is in particular relevant with respect to age. It is therefore that – where available – we use age-stratified data. In particular, this is available for France for all of the considered health outcomes and for Italy for deaths. Vaccine effectiveness appears to be similar across age groups, see Ref. 32. This is the case even though different age groups are more at risk than others. Vaccine effectiveness is defined as the ratio between the reduction in hazard rate due to vaccination and the initial hazard rate. Following your comment, we have added the following clarification in the text (Methods C). “Overall, vaccine effectiveness appears to be similar across age groups, and we have thus opted for a common estimate.³²”
3. Impact on the economy The authors estimate the effect of increases in vaccination rates on economic activity using a two-	_____

way fixed effect (TWFE) regression. This empirical approach seems justified, although the authors acknowledge a growing recent literature to address the limitations of TWFE. I have four main comments about this part of the manuscript:

a) The OECD weekly tracker seems like a great source of data. However, the effect relies on a debatable counterfactual based on a projected pre-pandemic growth trend. To ensure that the results are robust to this quite strong and crucial assumption, the authors should at least attempt an analysis – even at lower frequencies (e.g. monthly) – with a different indicator of economic activity. Alternatively, the result regarding the impact on economic activity should be clearly framed as a measurement of a very specific outcome, that is the OECD weekly tracker.

Following your suggestions, we have added robustness checks and applied the same regression as in Table 7 to alternative left-hand side variables: the Google Mobility Index and quarterly GDP series. Results have been added (see Table 8) which are consistent with the paper's main findings: a 10% increase in vaccination is associated with a 0.07*** increase in mobility, consistently with the intuition that by the second half of 2021, economic activity and mobility are only partially correlated as businesses and households have adapted to the pandemic (eg., with teleworking). The regression on quarterly GDP mimics the direct effect model (Table 7 column 3) and the coefficient associated with vaccination (0.03) is close to the one obtained from weekly regressions on the Weekly Tracker (0.04***).

We have added the following to the main text:
“and alternative dependent variables namely Google Maps mobility index and official quarterly GDP”

We have added the following paragraph in Methods D:

Measure of economic activity. The main analysis of the impact of vaccination on economic activity is based on the OECD Weekly Tracker, which proxies weekly GDP relative to the pre-crisis trend. Additional regression analyses were performed to ensure that the results are robust to the choice of an economic activity metric. Table 8 shows estimation results of regressions on both the Mobility Index and official quarterly GDP. The former is a high-frequency proxy for economic activity, although its relationship with GDP is not straightforward, and the latter is the official low-frequency measure of economic production. The regression on the Google Maps mobility index yields results that are comparable to the main regression (Table 7), although with a smaller direct effect of vaccination. From our preferred specification, an increase of 10 p.p. in vaccination, increases mobility by 6.9 p.p., which is an effect larger by a third than the effect on the Weekly Tracker. This seems consistent with the fact that mobility was only partially correlated with economic activity, as by the start of 2021 households and firms had both adapted to the pandemic constraints (through working from home, e-commerce, etc.). As a result, around two-thirds of the increase in mobility triggered by vaccination translates into an increase in economic activity. The regression on quarterly GDP year-on-two-year growth rates[1] cannot include regressors lagged by 4 weeks, which implies that the baseline model cannot be estimated. The direct effect model yields an estimate of the impact of vaccination (0.028) on GDP that is close to the one that was obtained from the regression on the Weekly Tracker (0.043), although it is much less precise given that the

	identification cannot rely on the high-frequency variations in both vaccination and economic activity. The results obtained from regressions on these two alternative measures confirm the robustness of the main effect captured with greater precision from the regression on the OECD Weekly Tracker.”
b) Similarly to the section on health outcomes, this level of granularity in the data fails to capture a potentially great heterogeneity in the effects across regions/areas with different economic/industrial structures. This should be better acknowledged in the paper.	We agree that both cross-country and cross-region heterogeneity can occur. The regression analysis in the main text focuses on the average effect. We added the following paragraph to better highlight this point and refer to the various robustness checks: “Results in Table 7 are to be understood as average effects; the regression analysis does not aim at capturing possibly heterogeneous effects across countries or time. The effect of vaccination on economic activity may differ across countries and regions, based on cross-country or cross-region differences in sectoral composition and institutional settings. Moreover, the effect of vaccination may also differ in time. This is the case as the protection granted by the vaccine may have a smaller impact when virus circulation is near zero. The present analysis simply aims at gauging the average impact of vaccination on the economy. Some of the sources of heterogeneity are uncovered in the robustness checks performed below.”
c)  ● The authors should make an effort in discussing the identifying assumptions underpinning their TWFE approach. ● Additionally, I believe that a proper discussion of the following elements is missing:  ○ how control and treatment countries are defined, what the actual treatment is, how timing of COVID certificate policies relates to differential treatment timing, ○ and how all this could affect estimation results. 	We are sorry that our exposition was not clear. For the economic estimates we do not apply a treatment / no-treatment country method. Instead we estimate the effect of vaccination on economic output (via the proxy of the tracker). In a second step, the economic effect of the COVID certificates are derived by combining the uplift on vaccination and the economic effect of vaccination. We make it clearer that we directly estimated the impact of vaccination of economic activity by adding a new paragraph at the start of section D. In order to clarify this to the reader, we now start “Methods D” with the following two paragraphs: “The analysis of the economic impact of COVID certificates is based on an indirect causal model: COVID certificates boost vaccination, and vaccination encourages individuals to resume economic activities, thus increasing GDP growth. The quantification exercise thus follows a two-step approach. First, we estimate the average impact of a marginal increase in vaccination rates on economic activity using two-way fixed-effect regressions based on a large panel of data from 46 countries. Second, the estimate of the average effect of vaccination on economic activity is combined with the estimate of the uplift in vaccination obtained in section B to gauge the effect of COVID certificates on economic activity. Average effect of vaccine uptake on economic activity. Our estimation is based on data from 46 countries. The identification exploits high-frequency

	within-country variations in vaccination rates, and assumes a static relationship whereby vaccination at time $t-28$ impacts economic activity at time t. We use a two-way fixed-effect regression and identify the effect through a difference-in-differences design, which assumes a common trend across countries conditional on several covariates described below (see also the descriptive statistics in Table 6).”
d) Related to point c), given the focus of the author’s on three countries (France, Italy, Germany), the use of a TWFE with no specific treatment/control may not be the best approach. Did the author consider – as in the first part of the work – estimating the economic impact using a synthetic control approach, looking for proper comparators for each country of interest?	The use of synthetic control for analysing the overall impact of vaccination on economic activity is made difficult by the relative synchronicity of vaccination across countries. It could have been possible to apply synthetic control to countries with early vaccination campaigns (the United States, United Kingdom, and Israel) but without a strong guarantee that the results might be generalised to the three countries of interest. That is the reason why we decided to assess the average impact of vaccination on economic activity using a large dataset of 46 countries and using high-frequency identification.
e) In section Methods D (page 18, below Table 4 [now Table 7], I encourage the authors to make the link between the counterfactual vaccine coverage obtained within the section on “Effect of COVID certificates on vaccine uptake” more explicit. The authors introduce their way to construct the counterfactual with an equation, but repeating this link in words would facilitate the reader.	Thanks for this suggestion. We added the following lines after Table 7: “The economic impact of COVID certificates is assessed through the indirect causal model whereby COVID certificates spur vaccination, and which in turn increases economic activity. As a result, the counterfactual for economic activity (i.e., in the absence of COVID certificates) is obtained by plugging the counterfactual for vaccination rates in the regression model. The latter provides an estimate for weekly GDP absent COVID certificates by subtracting the estimation of the impact of COVID certificates on weekly GDP, denoted by $\delta_{i,w}$, from the observed weekly GDP:”
f) In section Methods D or in an appendix, the manuscript would benefit of a table showing descriptive statistics for the variables used in the regressions in Table 4 [now Table 7].	As suggested, we added a table showing descriptive statistics, i.e., Table 6.
(i) Overall, the study provides a great overview of the potential effects of these policies. However, building on the previous lines, I would argue that the identification strategies used to study the effect on health outcomes and economic activity are not sufficiently strong to make causal claims. This does not diminish the value of the contribution, but calls for some rewording throughout the manuscript. (ii) Again building on my discussion above, the interpretation of the milder results for Germany could also be the result of a bias due to anticipation of stricter measures, since this country was the last of the three analysed to introduce COVID certificates. (iii) The article makes an appropriate use of references. The only suggestion that I can make is to	(i) Thank you for this comment. Following your suggestion, we have rephrased throughout the paper to qualify our results more carefully. For example, we added - (in the “Impact of COVID certificates on health outcomes” section) “We focus on the direct protection provided by vaccines, but omit the contribution of vaccines to reducing overall transmission and the fact that COVID certificates may alter epidemic dynamics through, for example, behaviour changes or different patterns of increasing natural immunity.” - (in the “Discussion” section) “Nevertheless, our analysis does not consider how COVID certificates may have altered epidemic dynamics and further study in this direction is required.” (ii) We agree that spillovers need to be considered. Please see our response to your comment 1c above and

engage more with the (albeit limited) macroeconomic literature on the effects of Covid shocks to support the discussion about potential mechanisms for the effect measured on economic activity. (iv) The abstract should be edited depending on changes to the rest of the manuscript, but is otherwise well written and self-contained. My only comment is that referencing in the abstract is likely not appropriate/necessary. (v) I hope I did not misinterpret anything, and do apologize if that is the case. Thanks again for this valuable piece of work.	the related additions to the manuscript. (iii) Following your suggestion, we have added the following sentence in the section “Impact of Covid certificates on the economy”: “The COVID-19 pandemic has had a large negative impact on the economy.” with the two references below 1. Baldwin R. & Weder di Mauro B. Economics in the Time of COVID-19. CEPR Press (2020). https://www.viet-studies.com/kinhte/COVID19_CPER.pdf 2. Chen S. et al. Tracking the economic impact of COVID-19 and mitigation policies in Europe and the United States. IMF Working Papers (2020). https://www.elibrary.imf.org/view/journals/001/2020/125/article-A001-en.xml (iv) Thank you, we have made small edits in the abstract to take into account the revision (integrating all reviewers’ comments). We have also removed referencing from the abstract, as suggested. (v) Thank you again for your valuable comments, which helped us improve the article.
---	--

Dear reviewer,

Thank you for appreciating our work and for your insightful comments and detailed suggestions to improve our article. We tried to address all your points detailed in the table below and have changed the manuscript accordingly.

We believe that the manuscript has improved and hope that you like the revised version.

Best wishes

The authors

NB. In our responses below, additional text is marked **bold** in case it is an addendum to an existing sentence that is stated in full for ease of readability.

Reviewer #2:	Response
Despite the recent shift of public attention away from COVID, the submitted paper's topic remains important and with future significance. The authors employ statistical and computational methods to estimate the impact of the COVID-19 mandates/certificates on vaccine uptake and extrapolate these results to evaluate the policies' impact on health outcomes and GDP. I view the paper as a useful contribution and complement to	Thank you for your careful reading and thoughtful comments. We have aimed to position our work more clearly throughout, and are addressing your comments below, in particular with a focus on your questions regarding innovation diffusion theory.

previous work on the same topic, specifically two existing prior studies: Mills and Ruttenauer (2021), hereafter MR and Karaivanov, Kim, Lu and Shigeoka (2021), hereafter KKLS, using (in part) different data sources and methodologies. The study is carefully and rigorously done and the results are in line (or larger in magnitude) from those in existing work, although the relevant literature and the paper's innovation and contribution to it is not sufficiently well-explained. My main concerns lie with the innovation diffusion model that the authors use to estimate the effect on vaccine uptake and the extrapolation of the results to health and economic outcomes. Detailed comments follow.	
Major comments:	
1a. I wonder whether the innovation diffusion model is the best way to think of COVID-19 vaccine uptake, especially in the early months. Vaccine eligibility was heavily regulated and proceeded in stages, based on age and other factors in most countries. The role of vaccine supply and eligibility should be clarified and explained, e.g., it is hard to believe the assumption of the model that adoption in January-April 2021 (even among 60+ adults) was purely based on a "coefficient of innovation" and "coefficient of imitation". How are the eligibility and supply constraints incorporated? At best the model parameters can be mechanically re-interpreted (the authors seem to fit from May onwards in the main results, please specify the start date) to proxy such constraints but this needs to be clearly stated (and its usefulness justified). In addition, discussion on the latter period, e.g., from June or July onwards when eligibility and supply constraints were essentially gone for first doses should be done in the context of the model (e.g., possibly use $K(t)$ instead of K on p.11 and in the estimation).	Thank you for raising this point. Indeed, supply & eligibility have been major drivers of vaccine uptake in the treatment countries. We therefore chose to separate the population into the two coarse age groups (over/under 60) and fit the latter only from May onwards. The reason why we did not further distinguish different groups is two-fold. First, data-availability. Second, and more convincingly, there is a theoretical literature that considers innovation diffusion processes in the presence of supply/eligibility constraints. In this paper [https://www.jstor.org/stable/183876] the authors show that the willingness to get vaccinated by the not-yet eligible (expressed by formal commitment or by a simple word-of-mouth) is an important driver for the adoption of the innovation among the eligible population. When the word-of-mouth effect equals the imitation effect, the resulting model boils down to the classic functional form of the Bass model that we used. In the text, we have now added the following clarification: “Regarding supply shortages and eligibility constraints, the early stages of the vaccine rollout were significantly impacted by supply constraints, which led most countries to give age-based priority. This effect is captured by an extension of the original innovation diffusion model,⁴⁴ but requires additional data – the fraction of the population willing to get vaccinated among the not-yet-eligible –, which is not available in our case. Next to the imitation and innovation coefficients, this model adds a ‘word-of-mouth’ coefficient that captures the influence of individuals willing to get vaccinated, though not yet eligible, on the eligible population. When the word-of-mouth coefficient is assumed equal to the imitation coefficient, this model with supply or eligibility constraints boils down to the original Bass model, thus lending further support to our modelling choice.”
1b. Related to point 1, on p.12 - the estimation for the 60 and older group begins in January and seems to assume all were eligible at the same time, is that correct? If not,	We added in Methods B: “Regarding supply shortages and eligibility constraints, note that the early stages of the vaccine rollout were significantly

how does it affect the results?	impacted by supply constraints, which led most countries to give age-based priority. An extension of the innovation diffusion model can capture this effect,⁴⁵ but requires additional data – the fraction of the population willing to get vaccinated among the not-yet-eligible –, which is not available in our case. Next to the imitation and innovation coefficients, this extended model adds a ‘word-of-mouth’ coefficient that captures the influence of willing, though not yet eligible, individuals on the eligible population. When the word-of-mouth coefficient is assumed equal to the imitation coefficient, the extended model with supply or eligibility constraints boils down to the original Bass model, thus lending further support to our modelling choice.” and: “The confidence intervals are narrow compared to those obtained in Fig. 1. This is the case as the fitting window is longer than for the whole population (due to eligibility), and, more importantly, it includes the three phases of the logistic curve (initial growth, inflection point, and deceleration). This leads to a precise fit, with narrow confidence intervals, and which is robust to shortening the fitting window by up to April.”
2. Technical questions related to the model (a) shouldn't p, the coefficient of innovation depend on risk-assessment in the context of COVID (e.g., current cases, deaths, etc.)? (b) what assumptions are being made about potential heterogeneity across people? (c) it is said on p.11 that the model assumes absence of major shocks but the Delta variant spread in the period of study which affects risk and hence potentially vaccine uptake. Need to discuss.	(a) Thank you for pointing this out. Note that, by adding a time-dependent effect (e.g., dependent on infection rates) one reverses to a statistical model such as OLS. Thus, this would depart from the innovation diffusion theory at the loss of its predictive power. (b) The innovation diffusion model is micro-founded: every individual has their own behaviour with regard to innovation and imitation. The model thus captures the heterogeneity of the population. (c) Thank you for pointing this out. It is true that the innovation diffusion model is not able to capture major exogenous shocks. In this regard, the second method, i.e., synthetic control, offers a robustness check addressing your concern. This is the case as synthetic control is robust to shocks that are common to all countries, such as the rapid spread of the (d) Delta variant. We added in “COVID certificates spur vaccination”: “As synthetic control is robust to shocks that are common to all countries, this suggests that around the period of the introduction of COVID certificates exogenous shocks, such as the rise of the Delta variant from late June, did not crucially pollute our estimates.”
3. If issues 1-2 cannot be addressed in a satisfactory way, the authors may consider making the synthetic control robustness check their main estimation method, however, that will diminish the contribution of the paper since MR use the same method.	We have opted to keep innovation diffusion theory as our main model for the reasons outlined above. In addition, note that synthetic control, while being used in MR, poses other problems as discussed in our paper (insufficient donor pool countries over time as most countries adopted similar measures, possible spillover effects).

4. When addressing health outcomes, e.g. averted ICU admissions on p.5, what other behavioural factors may exist that may offset the identified effect, e.g., higher uptake may mean engaging in more social contact or there may be selection effects? These factors can significantly affect the counterfactual ICU numbers estimated.	This is similar to the concern shared by reviewer 1. We are now aiming to more carefully qualify our analysis, for example, we added in the section “Impact of COVID certificates on health outcomes” the following sentence: “We focus on the direct protection provided by vaccines, but omit the contribution of vaccines to reducing overall transmission and the fact that COVID certificates may alter epidemic dynamics through, for example, behaviour changes or different patterns of increasing natural immunity.” Also, in the discussion part, we added: “Nevertheless, our analysis does not consider how COVID certificates may have altered epidemic dynamics and further study in this direction is required.”
5. a) How reliable are the "OECD weekly tracker" numbers based on "google trends search results and machine learning"? Are there any studies that have looked at that ex-post, relative to official national statistics? b) In particular, I assume these numbers come with standard errors or confidence bands but it appears that this uncertainty is not incorporated in the calculation of confidence intervals for the effect of increased vaccine uptake on GDP.	Thank you for these comments. a) The performance of the OECD Weekly Tracker was assessed for forecast simulations, see the OECD Working Paper (Woloszko, 2020), as well as the OECD Economic Outlook (OECD, 2020). The main results from this performance analysis are now added in Methods A: “The OECD Weekly Tracker’s accuracy was assessed using pseudo-real time forecast simulations.³⁹ On average across 46 countries over the period 2008Q1-2020Q4, it has a Root Mean Squared Error (RMSE) that is 17% lower than an autoregressive model that just uses lags of year-on-year GDP growth. The underlying model captures a sizeable share of business cycle variations, including the time around the global financial crisis (when the available data for training the algorithm was much smaller) and the euro area sovereign debt crisis. Its RMSE is on average 8% lower than an autoregressive model in 2008-10 and 41% lower in 2020. The timing of the downturn and subsequent rebound is well captured by the model, although the full magnitude of the negative shock in the second quarter of 2020 is typically under-estimated, given its unprecedented scale. The mean absolute error in predicting year-on-year GDP growth in the first (resp. second) quarter was 2.42 (resp. 3.86) p.p., compared with actual falls in GDP for the median country of 0.12% (resp. 10.4%). Note that the error is larger when the fall is very large, namely of a magnitude unseen before. The tracker thus provides a useful tool for real-time narrative analysis on a weekly basis, although it does not on average outperform models based on more standard variables, once these are eventually released.” Further, an analysis of the historic performance of the Tracker since the end of 2020 (which is not available yet) will be included in a forthcoming OECD Working Paper. The main takes are that the Tracker performed well during the COVID-19 crisis:  - the Median Absolute Error (MAE) across countries was high in Q2 2020 and lower after Q2 2020: around 1.35%pts (½ fall in the range [0.5 – 2.5]) - this corresponds to around 9% of the min-max range across countries (see Table below):

	MAE (%pts)	MAE (% min-max range)
2020 Q2	3.22	15%
2020 Q3	1.55	9%
2020 Q4	1.43	10%
2021 Q1	1.34*	9%*
2021 Q2	1.57*	8%*
2021 Q3	1.17*	7%*
2021 Q4**	1.14*	11%*

Note: For 2021, projections (*) are based on the counterfactual Tracker. Median MAE across 45 countries (**subject to data availability for 2021 Q4).

b) Thanks for pointing this out. Indeed, the Tracker is an estimate whose uncertainty needs to be taken into account when estimating the economic impact of COVID certificates. We have amended the methodology used to compute the confidence intervals of our estimates of the economic impact of COVID certificates. The new approach is described on p. 21-22 in Methods D. We use 300 bootstrap replicates of the Tracker, run the main regression 300 times, and applies Rubin's law to derive the empirical variance of the parameter of interest as the sum of a within component (the average of its variance across the 300 regressions) and a between component (the variance of beta across the 300 regressions). The resulting empirical variance estimate is used to compute the confidence intervals shown in Figure 3. The following has been added in the text (Methods D):

Estimating the impact of vaccination on economic activity needs to take into account the uncertainty relative to the measure of the latter, proxied by the OECD Weekly Tracker. The OECD Weekly Tracker is an algorithm prediction and includes an error term that can impact the accuracy of the estimation of the parameter β in the regression above. Its standard deviation thus needs to be adjusted to account for the uncertainty from both the econometric estimation and the weekly GDP measure. This can be achieved by using the multiple estimates from the OECD Weekly Tracker. The OECD provides 300 bootstrap series for the Tracker. Similar to the context of multiple imputation, we can apply Rubin's law (Rubin, 1987) to infer the variance of β from the estimates β_i obtained from each of the 300 replicates and using the following formula:

$$Var(\beta) = W_M + (1 + M^{-1})B_M$$

Where M is the number of replicate series ($M=300$ in our case), and W_M and B_M are respectively the within and between components, given by $W_M = M^{-1} \sum_{i=1}^M Var_i(\beta_i)$ and $B_M = (M - 1)^{-1} \sum_{i=1}^M [\beta_i - \beta]^2$. The total empirical standard deviation of β is 0.009, which results in 95% confidence intervals of [0.034, 0.070], which are around twice the size of the confidence intervals computed in Table 6.

6. Do the estimates on vaccine effectiveness in Table 3 [now Table 5] include medical evidence from the Delta variant? Given that the analysis ends in December 2021 I believe that this is important and hence a wider interval for these estimates may need to be considered. More specifically, are the numbers listed on top of p.16 the ones used in computing the counterfactuals, the beta(v)? In any case, it is warranted to include confidence intervals around these numbers. In general, how the CIs of the estimates are computed, what is included and what is not, should be made clearer.

Please note that this table is showing the distribution of the 4 different deployed vaccines in France, Germany, and Italy.

Regarding the effectiveness against severe outcomes (hospitalisation, ICU admission, or deaths), we have to rely on few studies from different contexts. First, we gathered evidence of the effectiveness of each type of vaccine in the context of a dominant Delta variant and accounting for the waning of immunity over time (References 32-36). Second, for each type of vaccine, we focused on the lower-bound estimates (i.e., the lower bound of the reported 95% CIs) to remain conservative.

Third, note that vaccine effectiveness is influenced by demographic differences regarding age, comorbidities, natural-acquired immunity, etc. As there is insufficient data for a statistical meta-study, we opted, given the available evidence, to select a fixed lower bound scenario for each vaccine type (to remain conservative), and then averaged over the vaccine distribution given in Table 5.

	Alternatively, one could perform a scenario-based analysis based on the estimates of the various studies (32-36). We believe that the choice of one single, conservative scenario is best for clarity, and most appropriate given the proximity of the various studies.
7. The GDP estimates appear to use standard OLS TWFE regressions which have been criticised in the recent methodological literature (as the authors correctly acknowledge on p.18). Methods and estimation routines for dynamic staggered interventions do exist now, e.g., in Stata there is 'csdid' by Callaway and Sant'Anna or eventstudyinteract by Sun and Abraham among other routines. This part of the paper would be improved by using an appropriate estimation method robust to heterogeneity and dynamic treatment effects across cohorts.	Thank you for this thoughtful comment. To respond we have expanded the paragraph in Methods D on robustness w.r.t. two-way fixed effect regressions: "Two-way fixed effects regressions. A recent literature⁵⁵⁻⁵⁷ has shed light on the limitations of two-way fixed effects regressions when the treatment effect is heterogeneous. Alternative estimators have been proposed,⁵⁵ which limit the risk of bias by restricting the comparisons between units and times. This literature is still young, and there are currently no satisfactory options for cases where the treatment is dynamic, with a fuzzy design and in the absence of stayers or quasi-stayers. More specifically, the fuzzy design (continuous treatment, with treatment intensities that vary both in time and across units) precludes the use of the estimators introduced by Callaway and Sant'Anna⁵⁸ or Sun and Abraham⁵⁹. In the absence of quasi-stayers, it is impossible to estimate the time fixed effects based on the imputation estimator introduced by Borusyak et al.⁶⁰ We assume that the effect is static, which means that the past treatments do not impact the outcome variable. The reason for this is mostly that the treatment is the cumulative vaccination rate. This implies a nested causal relationship so that it is unnecessary to model dynamic effects. Under the alternative assumption that the effect is dynamic, the DID_l estimator⁶¹ could be applied but our experiments were inconclusive. This seems to result from the fact that identification is based on the timing of the intervention rather than the intensity of the treatment; yet, in our paper, we focus on the intensity of the impact. Further, their DID_l estimator is a weighted average of DID between first-time switchers in t-l and not-yet-switchers in l. As a result, the estimation for each possible value of l is based on a small number of observations (e.g. 44 for l=1), and the results are not credible."
Minor comments	
1. p.2, top, the authors say: "The impact of COVID certificates on vaccine uptake, health outcomes, and the economy has not yet been investigated." This is (partially) incorrect - the authors need to engage more carefully the existing literature on the topic, specifically the MR and KKLS papers which have been publicly available since at least October 2021 and potentially other works that have appeared in the meantime.	We have rephrased this to: "COVID certificates have emerged during the pandemic as a new tool to spur vaccination uptake and they require further investigation."
2. p.5, the calculation of the impact on GDP appears to	Indeed, the calculation of the impact on GDP focuses on the average effect, using a linear regression. We assume that

assume a linear effect; this needs to be clarified/justified.	the average effect is a valid proxy of the effect around the implementation of the COVID certificates. As a robustness check, we computed time-varying estimates of the effect of vaccination on economic activity using causal machine learning, which provide similar orders of magnitudes. We have clarified this in section “Methods D” by adding the following sentence at the start of the paragraph “Time-varying relationship between vaccination and GDP”: “The regression analysis used to analyse the economic impact of vaccination yields an average effect which is assumed to be a valid approximation of the economic impact of the increment in vaccination caused by the COVID certificates. To test the robustness of this hypothesis, we use causal machine learning to produce a time-varying estimate of the economic effect of the vaccine.”
3. p.11, what is $f(t)$? Is it the same as $x(t)$?	Thank you for spotting this, it has been corrected. The fraction of vaccinated people is now denoted by $x(t)$ everywhere in the paper.
4. I couldn't find in the submitted materials the estimates and standard errors for the diffusion model parameters. Also, no detailed description of the estimation method was provided (is it simulated method of moments?) Please add details.	We have now included a table in Methods B with the parameters obtained in our estimates through mean-square minimisation; standard errors come from block bootstrap.
5. p.11 - what are the exact starting dates for the fitting exercise? For instance, if starting in April issues with supply and eligibility are important. Need to discuss how this may bias the estimates. At the very least robustness checks with later initial dates (e.g. June onwards) should be performed.	Vaccine uptake is primarily modelled via the innovation diffusion model, for which the fitting requires a significant time window to capture the curvature and inflection point of the adoption curve. Further, as argued above [responding to your major comment 1a], supply and eligibility constraints can be partially captured by this model as the willingness to get vaccinated acts as an accelerator to vaccination. For these reasons, we opted for a 100 day “fitting window” in the treated countries, that is, 100 days prior to the respective announcement of COVID certificates. Our estimates are robust to small changes in the fitting window. On the other hand, when using a much shorter window (e.g., 60 days) estimates become very noisy with respect to the exact window length. Following your suggestion, we have added the following explanation in “Methods B” section: “The fit is robust with respect to slightly longer or shorter fitting windows. On the other hand, when using a much shorter window (e.g., 60 days) estimates become very noisy with respect to the exact window length, likely because the fit does not appropriately pick up the curvature and inflection point of the logistic function.”
6. p.12, the calculation of counterfactual second doses seems to assume a 3-week gap. What is the actual average gap in the data? The assumed gap affects the health outcome estimates (e.g., may overestimate second doses), need to check robustness.	To evaluate the willingness to get a second shot, we measured the ratio of second doses at time $t+21$ over the number of first doses at time t. This ratio was not affected by the announcement of COVID certificates (as shown in Figure 7, added to Methods B). As mentioned earlier in the text, we used this ratio to build, from the counterfactual of first doses, a counterfactual for second doses.

Next to the Figure, we have added the following clarifications.

- (in section “Impact of COVID certificates on health outcomes”) “To estimate the impact of vaccine uptake on health outcomes, we construct counterfactuals for second-dose vaccine uptake **by assuming the same ratio between second and first dose uptake (with a three week lag) for the counterfactual and realised scenarios. (Methods B).**”
- (in Methods B) “Assuming the same ratio between second and first dose uptake (with a three-week lag, corresponding to the minimum required gap between first and second dose) for the counterfactual and realised scenarios is well-motivated, as this ratio was not affected by the intervention (see Fig. 7).

Figure 7. Ratio of second versus first vaccine doses around the date of the announcement of COVID certificates for France, Germany, and Italy.

7. p.17 - the use of the reproduction rate in the controls - how is it computed? It usually uses cases/deaths itself plus policy/behaviour, why include separately? Discuss. Also, if already control for cases, deaths etc. directly what would be any residual effect from temperature? Please explain.

The reproduction rate (source: Our World in Data) at time $t-28$ days (i.e., to account for the lag between the first dose and the vaccine full effectiveness) is an important control as it can influence both individual’s perception of current and future risks of contamination, and the virus circulation at time t which can in turn have an impact on economic activity at time t . Controlling for deaths and cases only at time $t-28$ may fail to account for the confounding effect of the past dynamic of virus circulation (i.e., ascendant or descendant, captured by the reproduction rate). Temperature is less important but remains a useful control, as temperature is auto-correlated, which implies that temperature at $t-28$ days is correlated with temperature at time t , which in turn has an impact on virus propagation and thus the economy at time t . It is true that the causal path between temperature at $t-28$ and vaccination at time $t-28$ is mostly blocked by variables relative to virus circulation at $t-28$. Including temperature thus provides an additional guarantee which comes at a near zero cost as the risk of over-parameterization is low thanks to the large sample size.

8. Why are the confidence intervals so narrow in Fig 7, compared to Fig. 1?

We have added the following comment in Methods B: "The confidence intervals are narrow compared to those obtained in Fig. 1. This is the case as the fitting window is longer than for the whole population (due to eligibility), and, more importantly, it includes the three phases of the logistic curve (initial growth, inflection point, and deceleration). This leads to a precise fit, with narrow confidence intervals, and which is robust to shortening the fitting window by up to

*References

Mills, M. and T. Ruttenauer (2021), "The effect of mandatory COVID-19 certificates on vaccine uptake: synthetic-control modelling of six countries", *Lancet Public Health* 7(1):E15-E22.

Karaivanov A., D. Kim, S. Lu, H. Shigeoka (2021), "COVID-19 Vaccination Mandates and Vaccine Uptake", preprint, <https://www.medrxiv.org/content/10.1101/2021.10.21.21265355v3>

Dear reviewer,

Thank you for appreciating our work and for your insightful comments and detailed suggestions to improve our article. We tried to address all your points detailed in the table below and have changed the manuscript accordingly.

We believe that the manuscript has improved and hope that you like the revised version.

Best wishes

The authors

NB. In our responses below, additional text is marked **bold** in case it is an addendum to an existing sentence that is stated in full for ease of readability.

Reviewer #3:	Response:
NCOMMS-22-02909: "The effect of COVID certificates on vaccine uptake, health outcomes, and the economy" This manuscript investigates the effect of mandatory vaccination certificates on vaccine uptake, health outcomes and the economy based on data from France, Germany and Italy. It uses innovation diffusion modelling to construct counterfactual outcomes. The results reveal a consistent impact of certificates on vaccine uptake, and a beneficial effect on health outcomes in France and Italy. The manuscript provides an important contribution to the literature. The analysis on vaccine uptake seems straightforward, also given the additional robustness checks. However, I have several comments on the other outcomes.	Dear reviewer, thank you for your positive remarks and thoughtful comments. We have addressed them in their entirety. We are detailing our changes in this table, and believe that the paper has substantially improved after this revision.

1. Significance This study is innovative, and it adds important findings to the literature on vaccine certificates. However, the statement “the incentive effect of COVID certificates on vaccine uptake, health outcomes, and the economy has not yet been investigated” (abstract and p.2, l.40) is not true. For instance (and as mentioned later on), Mills & Rüttenauer (2022) investigate the effect of certificates on vaccine uptake and on case rates in the supplement. The paper needs to clarify its additional contribution. Also, the working paper by Karaivanov et al. (2021) seems relevant.	Thank you for pointing this out; we have clarified it as follows. In the abstract, we have changed the sentence “While arguments for and against COVID certificates have focused on reducing transmission and ethical concerns, their effect on vaccine uptake, public health, and the economy has not yet been investigated.” with “[...] their effect on vaccine uptake, public health, and the economy requires further study.” Further, in the introduction we amended “As with other policy choices, the use of COVID certificates has often been questioned for ethical and political reasons,^{11,12} while advocates have focused on the potential to secure social interactions¹³ and only recently increased vaccine uptake has been considered.¹⁴⁻¹⁸” Finally, we also replaced “The impact of COVID certificates on vaccine uptake, health outcomes, and the economy has not yet been investigated.” with “COVID certificates have emerged during the pandemic as a new tool to spur vaccination uptake and they require further investigation.”
2. Validity of results / interpretation a. It should be clarified upfront that the analyses on additional outcomes (deaths, ICU admission, GDP) are not independent but are based on the results on vaccination uptake. Or do I misinterpret something here? b. It is stated several times that the paper finds robust effects of certification on health outcomes (e.g. p. 6, l. 216). However, the results on health outcomes in Germany are not statistically significant. The conclusions should clarify that (also in the abstract). c. I am concerned that the results on health outcomes might be confounded by the onset of omicron. The results (Figure 2) only indicate substantial differences between observed and the estimated counterfactual in December. However, according to Our World in Data, around one third of cases was already omicron in December (and I guess this is an under-estimation). This might drive the lower deaths and ICU rates in December, and it is difficult to say how much of it is attributable to the vaccination rate (Wolter et al. 2022).	a. Yes, our results on health outcomes and the economy are based on the results on vaccine uptake. This point has been clarified, thank you for pointing it out. See, for example, in the Abstract (“Further, this vaccine uptake averted”) and the Introduction (“We argue that the incentive effect of COVID certificates on vaccine uptake may be most critical and has averted adverse health and economic outcomes.”) b. Following your suggestion, we made the following changes: Abstract: “Varying government communication efforts and restrictions associated with COVID certificates may explain country differences, including the smaller and initially insignificant effect in Germany.” Discussion: “COVID certificates have had a sizable, robust positive effect on vaccination rates, health outcomes, and the economy in France, Germany (albeit only significantly towards the end of 2021), and Italy.” c. Thank you for this point, to which we can respond with the two following considerations:  (1) As you point out, Omicron started to increase throughout December, with ca. 0% at the beginning, <5% in mid-December and ca. 30% by the end of December in France, Germany, and Italy. (2) Regarding severe health outcomes (hospitalisation, ICU admission, deaths), they occur on average ca 14 days after infection.

	Thus, in particular, health hazards recorded by the end of the year, correspond to infections by mid December, when Delta was still clearly dominant. The effect of Omicron on our results is therefore negligible. Finally, we included the reference (Wolter et al. 2022) when mentioning the severity of Omicron.
3. Data a. Subsection A (p. 8) mentions data on vaccine acceptance. However, I could not find information where this is used in the analyses? b. Can you please give some information on the accuracy of the OECD weekly tracker? c. Could you clarify why the paper only looks at first doses in the main analysis? Could there be an additional effect on second doses?	a. Vaccine acceptance is used as a covariates in the synthetic control method. As already stated in Methods D, “covariates include annual GDP per capita, the average fatalities and cases over the pre-treatment period, the share of the population aged over 65, the average Mobility Index over 2020, and average vaccine acceptance over the first quarter of 2021”. b. [this comment is mirrored by review 2, for convenience we repeat the response here:] The performance of the OECD Weekly Tracker was assessed for forecast simulations, see the OECD Working Paper (Woloszko, 2020), as well as the OECD Economic Outlook (OECD, 2020). The main results from this performance analysis are now added in Methods A: “The OECD Weekly Tracker’s accuracy was assessed using pseudo-real time forecast simulations.³⁹ On average across 46 countries over the period 2008Q1-2020Q4, it has a Root Mean Squared Error (RMSE) that is 17% lower than an autoregressive model that just uses lags of year-on-year GDP growth. The underlying model captures a sizeable share of business cycle variations, including the time around the global financial crisis (when the available data for training the algorithm was much smaller) and the euro area sovereign debt crisis. Its RMSE is on average 8% lower than an autoregressive model in 2008-10 and 41% lower in 2020. The timing of the downturn and subsequent rebound is well captured by the model, although the full magnitude of the negative shock in the second quarter of 2020 is typically underestimated, given its unprecedented scale. The mean absolute error in predicting year-on-year GDP growth in the first (resp. second) quarter was 2.42 (resp. 3.86) p.p., compared with actual falls in GDP for the median country of 0.12% (resp. 10.4%). Note that the error is larger when the fall is very large, namely of a magnitude unseen before. The tracker thus provides a useful tool for real-time narrative analysis on a weekly basis, although it does not on average outperform models based on more standard variables, once these are eventually released.” Further, an analysis of the historic performance of the Tracker since the end of 2020 (which is not available yet) will be included in a forthcoming OECD Working Paper. The main takes are that the Tracker performed well during the COVID-19 crisis:  - The Median Absolute Error (MAE) across countries was high in Q2 2020 and lower after Q2 2020: around 1.35%pts (½ fall in the range [0.5 – 2.5]) - This corresponds to around 9% of the min-max range across countries

	MAE (%pts)	MAE (% min-max range)
2020 Q2	3.22	15%
2020 Q3	1.55	9%
2020 Q4	1.43	10%
2021 Q1	1.34*	9%*
2021 Q2	1.57*	8%*
2021 Q3	1.17*	7%*
2021 Q4**	1.14*	11%*

Note: For 2021, projections (*) are based on the counterfactual Tracker. Median MAE across 45 countries (**subject to data availability for 2021 Q4).

c. Thank you for raising this point, which relates to a similar remark from Reviewer 2. The following analysis, which is now included, shows that there was no effect beyond the additional first-dose uptake: the second dose uptake follows the same trend, but shifted in time.

To evaluate the willingness to get a second shot, we measured the ratio between the number of first doses at time t and the number of second doses at time $t+21$. This ratio was not affected by the announcement of COVID certificates. Following your comment, we have added the corresponding graph to Methods B (Figure 7). We therefore used this ratio to build a counterfactual for second doses from the counterfactual of first doses.

We have added the following clarifications.

- (in section "Impact of COVID certificates on health outcomes") "To estimate the impact of vaccine uptake on health outcomes, we construct counterfactuals for second-dose vaccine uptake **by assuming the same ratio between second and first dose uptake (with a three week lag) for the counterfactual and realised scenarios. (Methods B).**"

- (in Methods B) "Assuming the same ratio between second and first dose uptake (with a three-week lag, corresponding to the minimum required gap between first and second dose) for the counterfactual and realised scenarios is well-motivated, see Fig. 7."

4. Analytical approach

a. How does the statistical approach account for the fact that the countries have adopted other non-pharmaceutical measures, such as closing public venues and distancing rules, during the autumn of 2021? How do we know the differences in health outcomes between actual and estimated counterfactual health outcomes are due to the certificates and not due to the other measures? I see there is a stringency index included in the analysis on economic outcomes, but this seems similarly relevant for the analysis on health outcomes.

b. Similarly, how might the booster rollout in November and December affect the results on health outcomes? Again, it coincides with the period of the

a. We agree with you that COVID certificates, as well as other NPIs, have influenced the course of the pandemic. To isolate the effect of COVID certificates, we implicitly make the *ceteris paribus* assumption, that is, assuming all else, i.e., other NPIs such as mask mandates and social distancing, would have remained equal with/without the introduction of COVID certificates.

The stringency index is used (among other covariates) in the economic analysis to isolate the effect of vaccination on economic activity (GDP). On the other hand, regarding health outcomes, we refer to the statement above (*ceteris paribus*) and the fact that the average hazard rate from COVID only depends on vaccine effectiveness and vaccination rate.

b. The booster roll-out likely played a role in observed health outcomes. Yet, our analysis is concerned with the difference in the number of individuals who opted to start their 'vaccination program' after the introduction of

largest “treatment effect”. c. There is no information on how confidence intervals for Figure 2 / method C are calculated, or I am missing it. How do we account for the fact that the effect of vaccination on health outcomes as well as the counterfactual vaccination rates are estimates – both associated with insecurity? d. As far as I understand, the analysis of GDP (Figure 3) uses a single coefficient (beta_vaccinated from Table 4?) to estimate the counterfactual. This assumes that we have a homogeneous effect of vaccines on health outcomes. However, wouldn't we expect that this effect changes over time (e.g. declining marginal benefit with increasing overall vaccination rate, different effects because of omicron)? – Is there a way of making these estimates more flexible?	COVID certificates; none of them were eligible for a booster before the beginning of 2021. For example, in France, a person getting vaccinated on 12 July (the announcement date) could not get a booster until 2 January (due to a 3 week gap between first and second dose, and 5 months gap for the booster). To clarify this point we have rephrased in “Impact of COVID certificates on health outcomes” as follows: “Booster uptake does not factor in our model, as individuals who were not vaccinated before the announcement of the COVID certificate were not eligible to receive a booster during 2021”. And in Methods B: “We do not consider $v > 2$ doses as individuals who were not vaccinated before the announcement of the COVID certificate were not eligible to receive a booster before early 2022; a counterfactual is thus not needed. (For example, in France, a person getting vaccinated on 12 July –the announcement date– could not get a booster until 2 January due to a 3-week gap between first and second dose, and 5 months gap for the booster).” c. Regarding the effectiveness against severe outcomes (hospitalisation, ICU admission, or deaths), we have to rely on few studies from different contexts. As there is insufficient data for a statistical meta-study, we opted, given the available evidence, to select a fixed lower bound scenario for each vaccine type (to remain conservative), and then averaged over the vaccine distribution given in Table 5. Therefore, the CIs reported come only from the uncertainty of our estimation on vaccine uptake. We clarify this now in “Methods C”. d. Thank you for raising this point. Indeed, we also looked at a time-varying beta coefficient, and derived an alternative impact for the GDP. The two estimates being very similar, we have chosen to keep the simpler one, and report the other one in the Methods as a robustness check (see final Figure in Methods D).
5. Clarity and context The manuscript is clear and nice to read, sufficient context is provided. I have just two minor suggestions. a. Given the similarity to Mills & Rüttenauer (2022), also in terms of countries, it might be helpful for the reader to know how results compare to each other. b. On page 1, the paper is placed in the context of vaccine hesitancy and refusal. Do we know which groups are most responsive to certification? Is it those who oppose vaccines, or rather young people who feel no health-related needs to get vaccinated, or those who are previously undecided for instance?	a. Following your comment, we have added the following footnote ‘j’ (see below) in the section “COVID certificates spur vaccination”: “Mills and Rüttenauer¹⁴ estimate the effect of COVID certificates 20 days before and 40 days after their implementation, and find for France 12.8 (8.8–18.8) p.p.; for Germany 2.5 (-4.3–6.7) p.p.; and for Italy 6.6 (2.6–12.7) p.p., with a larger effect among the younger population. By contrast, our estimates start on the announcement dates and go until the end of 2021.” b. This is an interesting point, which was investigated by Klüver et al. (PNAS 2020), through a large-scale survey. Their results include 1) Undecided [to get a covid vaccine] are the most responsive to incentives. Among them the 20-40 years old are most responsive to freedom-related incentives such as COVID certificates,

	while 60+ are more responsive to other incentives, e.g., vaccination at local doctors; 2) Respondents who refuse to get vaccinated are also affected by the incentives, albeit considerably less. We also find that 18-59 are more responsive to Covid certificates than 60+ in France and Italy (where age-stratified data is available), but the additional uptake of 60+ has a bigger impact on health outcomes. We were already reporting estimates for the over 60 population at the end of the section. We now further mention: “In France and Italy – where age-dependent vaccine uptake statistics were available – we find that the impact was larger among the younger population.” “The results are in line with studies analysing the immediate period after the intervention in various countries using cross-country or state comparisons,^{14-18, j} and are consistent but overall more substantial than predicted by survey-based estimates.^{30”}
--	--

References

Mills, M. C., & Rüttenauer, T. (2022). The Effect of Mandatory COVID-19 Certificates on Vaccine Uptake: Synthetic-Control Modelling of Six Countries. *The Lancet Public Health*, 7(1), e15-e22. [https://doi.org/10.1016/S2468-2667\(21\)00273-5](https://doi.org/10.1016/S2468-2667(21)00273-5)

Karaivanov, A., Kim, D., Lu, S. E., & Shigeoka, H. (2021). COVID-19 Vaccination Mandates and Vaccine Uptake (Vol. 597). <https://doi.org/10.1101/2021.10.21.21265355>

Wolter, N., Jassat, W., Walaza, S., Welch, R., Moultrie, H., Groome, M., Amoako, D. G., Everatt, J., Bhiman, J. N., Scheepers, C., Tebeila, N., Chiwandire, N., Du Plessis, M., Govender, N., Ismail, A., Glass, A., Mlisana, K., Stevens, W., Treurnicht, F. K., . . . Cohen, C. (2022). Early assessment of the clinical severity of the SARS-CoV-2 omicron variant in South Africa: a data linkage study. *The Lancet*, 399(10323), 437–446. [https://doi.org/10.1016/S0140-6736\(22\)00017-4](https://doi.org/10.1016/S0140-6736(22)00017-4)

REVIEWERS' COMMENTS

Reviewer #1 (Remarks to the Author):

Dear authors,

thank you for your careful revision, and for your accurate account of changes to the paper in response to reviewers' comments.

All comments and suggestions were either incorporated or dismissed with a valid motivation. This work is timely, policy-relevant, and methodologically sound. Moreover, the current version acknowledges the assumptions and limitations of the proposed approach. My assessment is that this work is now ready for publication.

Thank you once again for the opportunity to review such an interesting piece of work.

Best regards.

Reviewer #2 (Remarks to the Author):

I appreciate the authors' work and detailed replies toward addressing my comments on the initial submission. While the revisions do address the majority of my previous comments, several important issues, mostly about the innovation diffusion model used in the estimation, are still not resolved in a satisfactory way and need further work and clarification, as suggested below. Below I use the same numbering as in my original report.

1. From reading the answers to my comments, I now understand better the trade-off faced by the authors in using the structural innovation diffusion model vs. the synthetic cohort method (SCM) to estimate the effect of the mandates on vaccine uptake and health outcomes. Essentially only short-term inference (up to some time in September) can be done using the SCM, since most control-group countries also introduced mandates in Fall 2021. Hence, to use all available data until the end of 2021 another approach is needed. This trade-off can be made clearer in the paper. (a) Given the above, to address my concerns about vaccine availability constraints and age-cohorting possibly affecting the estimates or interpretation thereof, the authors should clearly specify all assumptions made in the structural model including the "word of mouth" extension that allows such constraints to be incorporated. For example, in the 60-plus group, if 60% wanted to be vaccinated in January or February ("word of mouth" or "intent") but only 20% could because of supply constraints, how is this captured by the model parameters? In other words, if, at least initially the growth in % vaccinated is exogenous (relaxing the constraints) how does this map into the model and how are the parameter estimates to be (re-)interpreted?

1(b) Heterogeneity. It should be clearly stated that the only allowed heterogeneity is 60-plus vs below-60 and also by country but not within those categories.

2(a) - it should be mentioned as limitation that p is assumed constant over time (as well as other parameters). Then, the trade-off explained in the authors' response to my original comment should be included in the paper too.

2(b) I am afraid that the provided reply is not useful. It is not useful to say that every individual "has their own behaviour" - the parameters are the same for all and only ex-post heterogeneity is possible (some fraction adopt, some don't, up to given time t). It should be clearly stated that the model doesn't admit heterogeneity beyond the two different samples by age used for each country.

2(c) need to list the need to assume absence of shocks as model limitation and then say that it's partially addressed by the SCM robustness check.

6. (vaccine effectiveness) - please mention explicitly in the paper (e.g., using the wording from the response) that the estimates of VE used are lower bounds of the effects found in the medical literature.

Minor comments (using the same numbers from my original report)

1. Need to add brief discussion and numbers for the estimates in Karaivanov et al. (2021) to the new footnote j, the same way as you did for Mills and Ruttenauer. The footnote doesn't clearly say what the p.p. increase refers to (please edit the language). Also mention the difference in estimation methods used in the 3 papers (innovation diffusion, SCM, time series).

4. (model estimates and new Table 4). Please provide standard errors for the estimates. What does it mean (incl. in terms of the model equations) that the values of t_0 in Table 4 are negative? Is t_0 relative to something (what is $t=0$)? Please explain.

I am satisfied with how the rest of my comments have been addressed. Thank you.

Reviewer #3 (Remarks to the Author):

NCOMMS-22-02909A: "The effect of COVID certificates on vaccine uptake, health outcomes, and the economy"

Many thanks for thoroughly revising the manuscript. The authors have dealt with all my previous comments and have clarified and improved the manuscript. I have just one humble suggestion below.

In my view, the statements about the effect of the certificates on health outcomes (hospital admissions, deaths, and ICU patients) and GDP are still very strong given the assumptions made here. The abstract still reads as if those are independent causal estimates. Also the added sentence in the discussion (p. 6, l. 230: "Nevertheless, our analysis does not...") does not clearly state the limitations of these estimates. Given that the health estimates are solely based on the estimates and uncertainty bounds of the vaccine uptake analysis - and apart from that just assume a constant and fixed ratio of health outcomes between the vaccinated and the unvaccinated (although there have been changing variants, changing natural immunity, and different NPI levels) - it should be clearly stated that those are only indirect causal estimates. Currently, the reader has to carefully look for this explanation / limitation in the respective methods sections.

Reviewer 1:

Dear authors,
thank you for your careful revision, and for your accurate account of changes to the paper in response to reviewers' comments.

All comments and suggestions were either incorporated or dismissed with a valid motivation. This work is timely, policy-relevant, and methodologically sound. Moreover, the current version acknowledges the assumptions and limitations of the proposed approach. My assessment is that this work is now ready for publication.

Thank you once again for the opportunity to review such an interesting piece of work.

Best regards.

Dear reviewer,

Thank you for appreciating our revised manuscript. We are happy you like our work!

Best wishes

Reviewer 2:

Dear reviewer,

Thank you for your additional insightful comments and detailed suggestions to improve our article. We tried to address all your points detailed in the table below and have changed the manuscript accordingly. We believe that the manuscript has improved and hope that you like the revised version.

Best wishes

Reviewer #2 (Remarks to the Author):

I appreciate the authors' work and detailed replies toward addressing my comments on the initial submission. While the revisions do address the majority of my previous comments, several important issues, mostly about the innovation diffusion model used in the estimation, are still not resolved in a satisfactory way and need further work and clarification, as suggested below. Below I use the same numbering as in my original report.

1. From reading the answers to my comments, I now understand better the trade-off faced by the authors in using the structural innovation diffusion model vs. the synthetic cohort method (SCM) to estimate the effect of the mandates on vaccine uptake and health outcomes. Essentially only short-term inference (up to some time in September) can be done using the SCM, since most control-group countries also introduced mandates in Fall 2021. Hence, to use all available data until the end of 2021 another approach is needed. This trade-off can be made clearer in the paper.

*We amended the following sentence (changes in bold): "The method requires a sufficiently large control group, which becomes infeasible as more and more countries adopted COVID certificates **in fall 2021, hence our choice to use an alternative principal method.**"*

(a) Given the above, to address my concerns about vaccine availability constraints and age-cohorting possibly affecting the estimates or interpretation thereof, the authors should clearly specify all assumptions made in the structural model including the "word of mouth" extension that allows such constraints to be incorporated. For example, in the 60-plus group, if 60% wanted to be vaccinated in January or February ("word of mouth" or "intent") but only 20% could because of supply constraints, how is this captured by the model parameters? In other words, if, at least initially the growth in % vaccinated is exogenous (relaxing the constraints) how does this map into the model and how are the parameter estimates to be (re-)interpreted?

Given no data availability we are not able to estimate the extended model as explained in the corresponding section (we reworded it slightly this clearer). Importantly, we note that "when the word-of-mouth coefficient is assumed equal to the imitation coefficient, this model boils down to the original Bass model, thus lending further support to our modelling choice."

1(b) Heterogeneity. It should be clearly stated that the only allowed heterogeneity is 60-plus vs below-60 and also by country but not within those categories.

*We have made this point clearer in the main text, by amending the following sentence. "Further, we estimate the effect of COVID certificates on vaccine uptake ~~among the older population~~ **by splitting the population into over and under 60 years of age.**" We believe that the country heterogeneity is clear from context.*

2(a) - it should be mentioned as a limitation that p is assumed constant over time (as well as other parameters). Then, the trade-off explained in the authors' response to my original comment should be included in the paper too.

We added the following sentence on your suggestion: "Innovation diffusion theory assumes constant parameters, which may be seen as a limitation. On the other hand, adding a time-dependent effect would result in a statistical model such as ordinary least squares (OLS), which does not have predictive power."

2(b) I am afraid that the provided reply is not useful. It is not useful to say that every individual "has their own behaviour" - the parameters are the same for all and only ex-post heterogeneity is possible (some fraction adopt, some don't, up to given time t). It should be clearly stated that the model doesn't admit heterogeneity beyond the two different samples by age used for each country.

Innovation diffusion theory is micro-founded in the following sense: Every individual has their own, heterogeneous, threshold at which they decide to adopt the innovation. The theory posits that these thresholds follow a logistic distribution. Therefore, the heterogeneity regarding countries or age groups assumes different parameters of the logistic distribution for different populations.

*We amended the following sentence as follows: "Innovation diffusion theory attempts to formalise the way in which an innovation is gradually taken up by a population, where early adopters are then joined by followers. **Every individual has their own, heterogeneous, threshold at which they decide to adopt the innovation.** The model relies on growth models with capacity limits, i.e., logistic curves, **positing that thresholds are distributed accordingly.** In our context, vaccines are the innovation that every (eligible) person may choose to adopt.*

2(c) need to list the need to assume absence of shocks as model limitation and then say that it's partially addressed by the SCM robustness check.

*We have amended in the main text in the Section "COVID certificates spur vaccination" [change in bold]: "This micro-founded model has been widely used due to its tractability and interpretability,²⁹ is robust to capacity constraints, **but requires the assumption of no exogeneous shocks.**"*

*"As synthetic control is robust to shocks that are common to all countries, this suggests that around the period of the introduction of COVID certificates exogeneous shocks, such as the rise of the Delta variant from late June, did not crucially pollute our estimates. **Thus, their exclusion in the innovation diffusion model is appropriate.**"*

6. (vaccine effectiveness) - please mention explicitly in the paper (e.g., using the wording from the response) that the estimates of VE used are lower bounds of the effects found in the medical literature.

In Section “Impact of COVID certificates on health outcomes” we added “by considering lower bounds from the medical literature”.

In Methods C we added:

“We rely on the few available studies from different contexts, thus precluding a statistical meta-study. We therefore gather conservative, lower-bound estimates below.”

Minor comments (using the same numbers from my original report)

1. Need to add brief discussion and numbers for the estimates in Karaivanov et al. (2021) to the new footnote j, the same way as you did for Mills and Ruttenauer. The footnote doesn't clearly say what the p.p. increase refers to (please edit the language). Also mention the difference in estimation methods used in the 3 papers (innovation diffusion, SCM, time series).

We have made the following amendments as suggested:

“Our results are consistent but overall more substantial than predicted by survey-based estimates,²² and are in line with studies analysing the immediate period after the intervention in various countries using cross-country or state comparisons using different methods.¹⁵⁻¹⁹ In particular, using synthetic control, Mills and Rüttenauer¹⁵ estimate the effect of COVID certificates on vaccine uptake 20 days before (as a proxy for the date of their announcement) and 40 days after their implementation at 12.8 (8.8–18.8) p.p. of the entire population for France, 2.5 (-4.3–6.7) p.p. for Germany, and 6.6 (2.6–12.7) p.p. for Italy, with a larger effect among the younger population. Using time series methods, Karaivanov et al.¹⁶ estimate the effect of COVID certificates on vaccine uptake from their announcement until 31 October 2021 at 8 p.p. of the entire population for France, 4.7 p.p. for Germany, and 12.1 p.p. for Italy. By contrast, we estimate the effect from the announcement dates until the end of 2021 using innovation diffusion theory.”

Also, we moved this into a paragraph, as we were instructed to remove all footnotes from the manuscript.

4. (model estimates and new Table 4). Please provide standard errors for the estimates. What does it mean (incl. in terms of the model equations) that the values of t_0 in Table 4 are negative? Is t_0 relative to something (what is $t=0$)? Please explain.

Thank you for spotting this. We have added the confidence intervals. Also we clarify t_0 : “For convenience, we define t_0 to be the number of days before or after 100 days prior to the announcement of the COVID certificate. Note that t_0 is not decisive for the estimation, as initial growth of the logistic function is near zero.”

I am satisfied with how the rest of my comments have been addressed. Thank you.

Reviewer 3:

Many thanks for thoroughly revising the manuscript. The authors have dealt with all my previous comments and have clarified and improved the manuscript. I have just one humble suggestion below.

In my view, the statements about the effect of the certificates on health outcomes (hospital admissions, deaths, and ICU patients) and GDP are still very strong given the assumptions made here. The abstract still reads as if those are independent causal estimates. Also the added sentence in the discussion (p. 6, l. 230: "Nevertheless, our analysis does not...") does not clearly state the limitations of these estimates. Given that the health estimates are solely based on the estimates and uncertainty bounds of the vaccine uptake analysis - and apart from that just assume a constant and fixed ratio of health outcomes between the vaccinated and the unvaccinated (although there have been changing variants, changing natural immunity, and different NPI levels) - it should be clearly stated that those are only indirect causal estimates. Currently, the reader has to carefully look for this explanation / limitation in the respective methods sections.

Dear reviewer,

Thank you for your additional insightful comments and detailed suggestions to improve our article. We tried to address your remaining concern and have changed the manuscript accordingly. We believe that the manuscript has improved and hope that you like the revised version.

Best wishes

We replaced in the abstract "Further, this vaccine uptake averted ..." by "Based on these estimates ...".

*In the discussion we amended in bold: "Nevertheless, **our analysis relies on the estimated increase in vaccine uptake** and does not consider how COVID certificates may have altered epidemic dynamics **as well as influenced other policy choices.**"*